# Emergence of Dip2-mediated specific DAG-based PKC signalling axis in eukaryotes

**Sakshi Shambhavi[1,2†], Sudipta Mondal[1†], Arnab Chakraborty[3], Nikita Shukla[1], Bapin Kumar Panda[1], Santhosh Kumar[1,2], Priyadarshan Kinatukara[1], Biswajit Pal[1], Siddhesh S Kamat[3], Rajan Sankaranarayanan[1,2*]**

[1]CSIR-Centre for Cellular and Molecular Biology, Hyderabad, India; [2]Academy of Scientific and Innovative Research (AcSIR), Ghaziabad, India; [3]Department of Biology, Indian Institute of Science Education and Research (IISER), Pune, India

## eLife Assessment

This is an interesting study that adds **useful** new data addressing how different DAG pools influence cellular signaling. The study dissects how the enzyme Dip2 modulates the minor lipid signaling DAG pool, which is distinct from the lipid metabolism DAG pool utilized in membrane production. Overall the analysis is **solid** and broadly supports the claims.

**\*For correspondence:**
sankar@ccmb.res.in

[†]These authors contributed equally to this work

**Abstract** Diacylglycerols (DAGs) are used for metabolic purposes and are tightly regulated secondary lipid messengers in eukaryotes. DAG subspecies with different fatty-acyl chains are proposed to be involved in the activation of distinct PKC isoforms, resulting in diverse physiological outcomes. However, the molecular players and the regulatory origin for fine-tuning the PKC pathway are unknown. Here, we show that Dip2, a conserved DAG regulator across Fungi and Animalia, has emerged as a modulator of PKC signalling in yeast. Dip2 maintains the level of a specific DAG subpopulation, required for the activation of PKC-mediated cell wall integrity pathway. Interestingly, the canonical DAG-metabolism pathways, being promiscuous, are decoupled from PKC signalling. We demonstrate that these DAG subspecies are sourced from a phosphatidylinositol pool generated by the acyl-chain remodelling pathway. Furthermore, we provide insights into the intimate coevolutionary relationship between the regulator (Dip2) and the effector (PKC) of DAG-based signalling. Hence, our study underscores the establishment of Dip2-PKC axis about 1.2 billion years ago in Opisthokonta, which marks the rooting of the first specific DAG-based signalling module of eukaryotes.

## Introduction

Diacylglycerol (DAG) is a conserved lipid molecule with well-established roles in membrane lipid biogenesis and storage lipid production in all life forms. DAGs are synthesised via de novo pathways of lipid metabolism utilising fatty acid precursors, while it can also be generated as a by-product of multiple salvage pathways in eukaryotes (*Carrasco and Mérida, 2007*; *Eichmann and Lass, 2015*). DAGs are unique as they lack headgroups, unlike the other abundant lipid classes. They exhibit diversity due to variations in acyl-chain length and unsaturation. These acyl-chain variations are proposed to be the basis for the multifaceted physiological roles of DAGs in metabolic and signal transduction pathways (*Marignani et al., 1996*; *Milne et al., 2008*; *Schuhmacher et al., 2020*). The signalling function is executed by various 'DAG effector' proteins such as DAG-dependent protein kinases,

Chimaerin, Munc13, RasGRP, etc. involved in cell-cycle progression, neurotransmitter release, actin cytoskeleton organisation, and malignant transformation (*Colón-González and Kazanietz, 2006*; *Yang and Kazanietz, 2003*). Studies conducted ~four decades ago have indicated the role of acyl-chain compositions of DAGs in Protein Kinase C (PKC) activation (*Kishimoto et al., 1980*; *Marignani et al., 1996*; *Mori et al., 1982*; *Kamiya et al., 2016*; *Masayoshi et al., 1987*). However, the cellular pathways for such specific acyl-chain-based regulation of signalling events have not been identified till date.

PKC, a serine/threonine protein kinase, is a classic example of a eukaryotic DAG-based signalling molecule. Mammalian PKC isoforms play a key role in a myriad of biological processes like cell differentiation, cell migration, mitochondrial functioning, cell polarity, etc. (*Newton, 2018*) and its dysregulation is associated with various pathological conditions such as cancer, diabetes, Alzheimer's, heart disease, obesity, and age-related metabolic disorders (*Etcheberrigaray et al., 2004*; *Marrocco et al., 2019*; *Nishizuka, 1984*; *Schmitz-Peiffer and Biden, 2008*). Fungi, on the other hand, harbour a single PKC gene (*PKC1*) that is implicated in regulating the cell wall biosynthesis pathway, rationalising its essentiality under normal growth conditions (*Levin, 2005*). Genetic or pharmacological impairment of Pkc1 in pathogenic fungi like *Candida albicans, Cryptococcus neoformans etc.,* confers hypersensitivity to various antifungal drugs and drastically attenuates its proliferation and virulence in murine model (*Heinisch and Rodicio, 2018*; *LaFayette et al., 2010*) suggesting a crucial role of Pkc1 in pathogenicity of diverse fungal groups. The array of pathologies and disrupted cellular processes owing to hampered Pkc1 signalling warrants the necessity of stringent regulation of Pkc1 activity.

Although DAG-based activation of PKC is well established in metazoans, the role of specific acyl-chain compositions in fungi is still a mystery. Additionally, the molecular players regulating these signalling DAGs remain underexplored. Recently, we have identified and functionally characterised a conserved protein family, Disco-Interacting Protein 2 (Dip2) (earlier annotated as Cmr2 in yeast; SGD ID: S000005619), harbouring two tandem fatty acyl-AMP ligase-like domains (FLD1 and FLD2) along with a DMAP-binding domain 1 (DBD1) at the N-terminus (*Mondal et al., 2022*). We showed that Dip2 regulates specific DAG subspecies in terms of its acyl chain length and unsaturation across fungi and animals. Dip2 from *Saccharomyces cerevisiae* has been shown to be involved in converting C36:0 (18:0/18:0) and C36:1 (18:0/18:1) DAGs to triacylglycerols (TAGs) with corresponding chain lengths. These DAG subspecies constitute only ~0.5% of the total lipid pool but are important for maintaining the organellar and thereby cellular homeostasis. However, the necessity to introduce such strict regulation of specific DAG subspecies in Opisthokonta remains elusive.

Here, we show that Dip2 has emerged as a modulator of Pkc1 signalling in the model yeast *Saccharomyces cerevisiae* by regulating the chain-length specific DAGs. By combining genetic and lipidomic approaches, we have demonstrated that the absence of Dip2, which leads to the accumulation of C36:0 and C36:1 DAGs, results in hyperactivation of Pkc1. This, in turn, elevates the downstream signalling of cell wall integrity (CWI) pathway leading to cell wall stress resistance. On the contrary, the bulk DAG pool, which is generally acted upon by canonical DAG metabolising enzymes, fails to activate Pkc1. The DAG binding domain of Pkc1 displays a remarkable specificity to certain DAG subspecies in vitro, reflecting the in vivo specificity. To understand this metabolic segregation of DAG pool, we have tracked the source of the Dip2-metabolised DAG subspecies involved in Pkc1 activation. Interestingly, it was found to be channelled majorly through the phosphatidylinositol (PI) pool enriched with C18:0 fatty acyl chain generated by acyl-chain remodelling process. Furthermore, an extensive phylogenetic correlation analysis has revealed that Dip2 and PKC co-emerged and co-evolved during early Opisthokonta evolution to establish the selective DAG-based PKC signalling axis.

## Results

### Dip2 is involved in the regulation of Pkc1-mediated cell wall integrity pathway

Initial phenotypic screening revealed that Dip2 is necessary for the survival of yeast in various stress conditions like ER stress, osmotic stress, etc. (*Mondal et al., 2022*). However, *Dip2* knockout yeast (hereafter referred to as *Δdip2*) outperforms the wildtype (*WT*) strain under cell wall stress conditions (*Mondal et al., 2022*). This suggested a possible link between Dip2 and the cell wall integrity (CWI) pathway of yeast, which is governed by Pkc1 and the downstream mitogen-activated protein kinase

(MAPK) signalling cascade. Given the requirement of DAGs for Pkc1 activation and the role of Dip2 in regulating DAG subspecies, we hypothesised that Dip2 maintains optimal levels of DAG subspecies necessary for activation of CWI pathway (*Figure 1A*). To test this, we compared the fitness of *WT* and *Δdip2* yeast in the presence of cell wall stress-inducing agents like Congo red (CR) and Calcofluor white (CFW) (*Roncero and Durán, 1985*), using serial dilution assay in synthetic complete (SC) media (*Figure 1B*) as well as in rich media (*Figure 1—figure supplement 1*). The two biological replicates, *Δdip2*_Colony1 and *Δdip2*_Colony2 showed increased resistance to cell wall stress, and the pheno-type was restored through genetic complementation with Dip2 under its native promoter. Additionally, we employed colony forming unit (CFU) assay for *WT* and *Δdip2* under cell wall stress. Counting the CFU also showed a similar and significant difference in the growth of *WT* and *Δdip2* colonies in the presence of CR (*Figure 1C*). These observations suggest the regulatory role of Dip2 in CWI pathway of yeast. Since cell wall stress is associated with Pkc1-mediated activation of CWI pathway, we sought to test the status of Pkc1 signalling and its probable link to Dip2-regulated DAG subspecies. First, we probed the phosphorylation levels of a downstream MAPK effector of Pkc1 pathway, i.e., Slt2 (Suppressor of the LyTic phenotype) (*Lee et al., 1993*; *Levin, 2005*) to quantitatively assess the extent of Pkc1 activation. In agreement to the CWI phenotype, a ~ threefold increase in phosphorylation of Slt2 (pSlt2) was observed in *Δdip2* as compared to *WT*, where *WT* treated with CFW is used as a positive control for pSlt2 increase (*Figure 1D*).

Next, we sought to understand if the hyperactivation of Pkc1 in *Δdip2* is DAG dependent and involves the catalytic activity of Dip2, i.e., conversion of selective DAGs to TAGs. To answer this, we generated two catalytic mutants of Dip2 (Dip2$^{D523A}$ and Dip2$^{L687A}$), that abolish its adenylation activity and have been shown to be unable to decrease the selective DAG levels on complementing *Δdip2*, as reported in our previous study (*Mondal et al., 2022*). The mutants were cloned under its native promoter in pYSS01 plasmid and expressed in *Δdip2* strain (*Figure 1—figure supplement 2A*). The mutants could not diminish the cell wall stress resistance in *Δdip2*, unlike wild-type Dip2 (*Figure 1E*). Additionally, complementation with the catalytically inactive Dip2 could not bring back the pSlt2 level in *Δdip2* (*Figure 1—figure supplement 2B*), suggesting a direct correlation between the enzymatic activity of Dip2 and Pkc1 hyperactivation. Hence, the catalytic activity of Dip2, i.e., the regulation of DAG subspecies level is essential in maintaining the active pool of Pkc1 for optimal CWI signalling.

Additionally, to confirm that the cell wall stress resistance observed in *Δdip2* is Pkc1 mediated, we inhibited Pkc1's activity using cercosporamide, a selective Pkc1 inhibitor (*Sussman et al., 2004*) and observed decrease in the resistance to cell wall stress (*Figure 1F*). The increase in pSlt2 level in *Δdip2* was also brought back to *WT* level upon treatment with cercosporamide (*Figure 1G*). This implies that the hyperactivation of Pkc1 in *Δdip2* causes the cell wall stress resistance.

These observations intrigued us to ask how Dip2 spatially regulates Pkc1 as both reside at different subcellular locations inside the cell. Since Pkc1 is a bud site and plasma membrane localising protein (Andrews P. and Stark M., 2000), we wondered whether Dip2 translocates to plasma membrane under cell wall stress. To examine this, we used *WT* strain with endogenous Dip2 tagged with GFP and subjected it to cell wall stress. As Dip2 predominantly localises to the mitochondria, we co-stained cells with Mitotracker Red and found that there is no change in the spatial organisation of Dip2 (*Figure 1—figure supplement 3*). This suggests that Dip2 and Pkc1 could be cross talking through other ways that are yet to be explored.

## Dip2-regulated DAG subspecies activate Pkc1 signalling via selective interaction

Since Dip2 is known to regulate the levels of C36:0 and C36:1 DAGs in yeast, we asked whether these specific DAG subspecies play any role in activation of Pkc1 signalling. First, we examined how the system responds to cell wall stress in terms of the DAG levels. While tracking the Pkc1 signalling at temporal scale, we observed profound activation of Pkc1 pathway after 20-30 min of CFW treatment (Fig. 2—fig. supplement 1A). Hence, we performed the lipidomic analysis of *WT* yeast pulse-treated with CFW for 30 min. Interestingly, the CFW treatment resulted in several fold increase in selective DAGs which normally constitute only 4-5% of the total DAG pool. We observed a ~eightfold increase in C36:0 DAGs and ~sixfold increase in C36:1 DAG subspecies levels compared to the untreated *WT* sample, while the other DAG species levels remained unaffected (Fig. 2 A). Thus, the data clearly indicates that Pkc1 activation in yeast might require C36:0 and C36:1 DAG subspecies. We have also

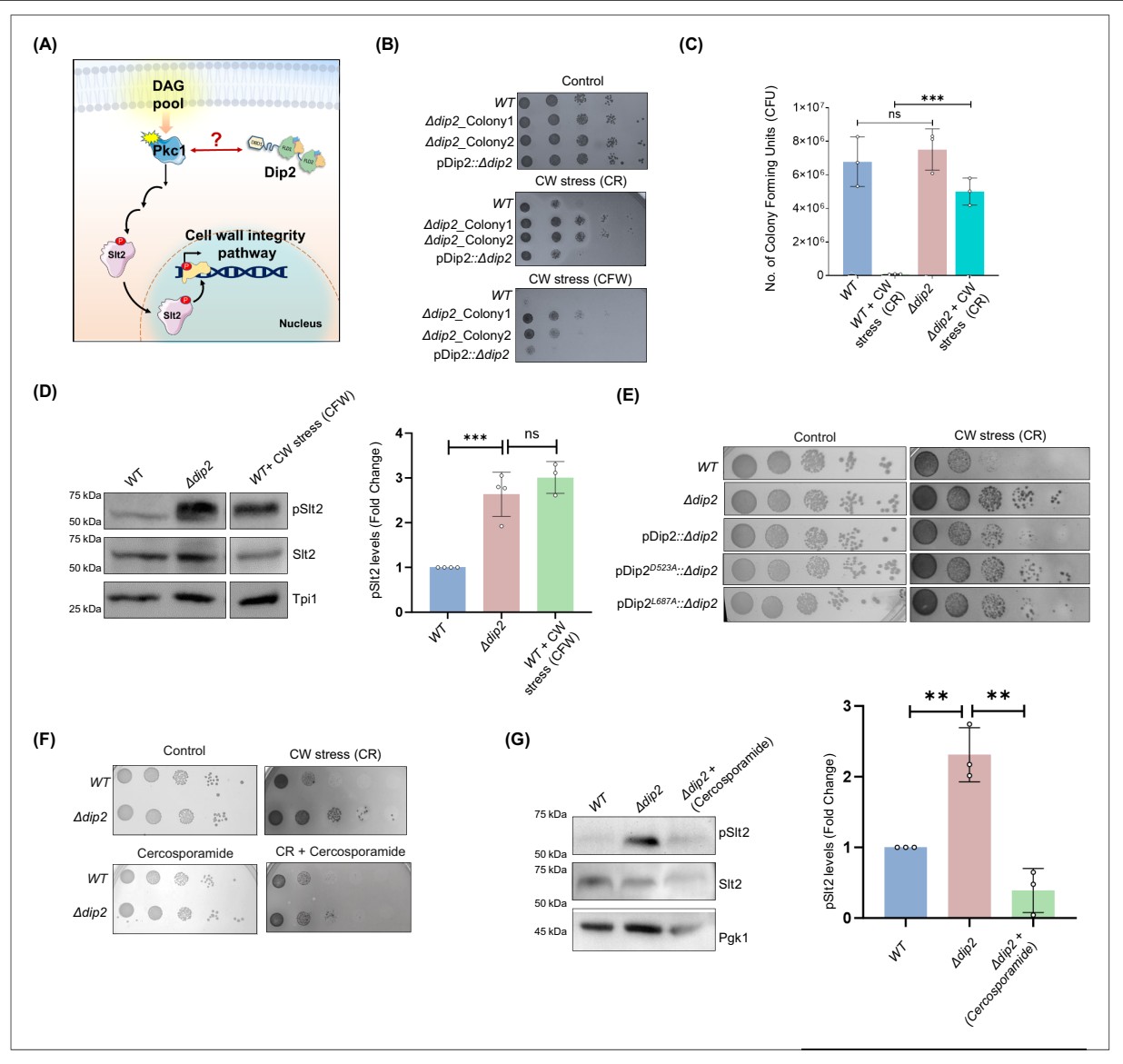

**Figure 1.** Deletion of disco-interacting protein 2 (Dip2) leads to increased cell wall stress resistance. (**A**) A model depicting the possible role of Dip2 in regulating diacylglycerol (DAG) species required for protein kinase C (PKC) activation which governs the cell wall integrity (CWI) pathway in yeast by activating the downstream mitogen-activated protein kinase (MAPK) cascade. Activation of the MAPK cascade results in increased cell wall synthesis, thereby strengthening the cell wall. Dip2 has been depicted in its three-domain architecture, harbouring DMAP-binding domain 1 (DBD1), tandem fatty acyl-AMP ligase-like domains (FLD1 and FLD2). (**B**) Serial dilution assay for wild-type (*WT*), two biological replicates of *Δdip2* (*Δdip2_*Colony1 and *Δdip2_*Colony2; *Δdip2_*Colony1 has been used for further experiments). pDip2::*Δdip2* represents *Δdip2_*Colony1 complemented with Dip2 expressed under its native promoter. Control plate contains synthetic complete (SC) media and cell wall (CW) stress plate has either congo red (CR) (100 µg/mL) or calcofluor white (CFW) (50 µg/mL). N=3. (**C**) Colony forming units of *WT* and *Δdip2* grown in SC media control and cell wall stress induced by CR (100 µg/mL). Data are represented as mean ± SD (unpaired, two-tailed Student's t-test; n=3; ***p<0.001; ns = not significant). N=3. (**D**) Representative western blot showing pSlt2 (56 kDa) levels in *WT* and *Δdip2* compared to the *WT* treated with CFW for 30 min, used as a positive control. Bar graph showing quantification of fold change in the pSlt2 levels, normalised with total Slt2. Triose phosphate isomerase (Tpi1) (27 kDa) has been used as a loading control. Data are represented as mean ± SD (unpaired, two-tailed Student's t-test; n>3; ***p<0.001; ns = not significant). (**E**) Serial dilution assay image in synthetic complete (SC) media control plate and in the presence of Congo red (100 µg/mL) for *WT* and *Δdip2* compared with *Δdip2* complemented with wildtype and catalytically inactive Dip2 mutants (Dip2^D523A^ and Dip2^L687A^), expressed in a plasmid pYSM7, under its native promoter. N=3. (**F**) Serial dilution assay for *WT* and *Δdip2* grown in SD media control and in the presence of CR (100 µg/mL) and cercosporamide (2 µg/mL). N=3. (**G**) Representative western blot for *WT*, *Δdip2,* and cercosporamide treated *Δdip2* showing the levels of pslt2 (56 kDa). Quantification of pSlt2 levels for *Δdip2* treated with cercosporamide (5 µg/mL) compared to *WT* and *Δdip2* control, normalised with total Slt2. Phosphoglycerate kinase (Pgk1) (45 kDa) has been used as a loading control. Data are represented as mean ± SD (unpaired, two-tailed Student's t-test; N=3; **p<0.01; ns = not significant).

*Figure 1 continued on next page*

*Figure 1 continued*

The online version of this article includes the following source data and figure supplement(s) for figure 1:

**Source data 1.** Spot assay for WT, *Δdip2,* and pDip2::*Δdip2* under cell wall stress.

**Source data 2.** Spot assay for WT, *Δdip2,* and pDip2::*Δdip2* under cell wall stress.

**Source data 3.** Quantification of colony forming unit (CFU) for WT and *Δdip2* in the presence of CR.

**Source data 4.** Western blot and quantification for wild-type (WT), *Δdip2,* and WT treated with calcofluor white (CFW).

**Source data 5.** Western blot and quantification for wild-type (WT), *Δdip2,* and WT treated with calcofluor white (CFW).

**Source data 6.** Spot assay for the catalytic mutants of disco-interacting protein 2 (Dip2).

**Source data 7.** Spot assay for the catalytic mutants of disco-interacting protein 2 (Dip2).

**Source data 8.** Spot assay for wild-type (WT), *Δdip2* in the presence of Congo red (CR) and cercosporamide.

**Source data 9.** Spot assay for wild-type (WT), *Δdip2* in the presence of Congo red (CR) and cercosporamide.

**Source data 10.** Western blot and quantification for wild-type (WT), *Δdip2,* and *Δdip2* treated with cercosporamide.

**Source data 11.** Western blot and quantification for wild-type (WT), *Δdip2,* and *Δdip2* treated with cercosporamide.

**Figure supplement 1.** Serial dilution assay in YPD media for wild-type (WT), *Δdip2,* pDip2::*Δdip2* represents *Δdip2* complemented with disco-interacting protein 2 (Dip2) expressed under its native promoter.

**Figure supplement 1—source data 1.** PDF file containing original spot assay plate images for *Figure 1—figure supplement 1*, indicating the relevant spots and treatments.

**Figure supplement 1—source data 2.** Original files for spot assay plate images displayed in *Figure 1—figure supplement 1*.

**Figure supplement 2.** Dip2-GFP mutant expression and pSlt2 quantification.

**Figure supplement 2—source data 1.** PDF file containing original western blots for *Figure 1—figure supplement 2A*, indicating the relevant bands.

**Figure supplement 2—source data 2.** Original files for western blot analysis displayed in *Figure 4F*.

**Figure supplement 2—source data 3.** PDF file containing original western blots for *Figure 1—figure supplement 2B*, indicating the relevant bands.

**Figure supplement 2—source data 4.** Original files for western blot analysis displayed in *Figure 1—figure supplement 2B*.

**Figure supplement 3.** Localization of Dip2-GFP under cell wall stress condition.

**Figure supplement 3—source data 1.** Original microscopy image for GFP and mCherry in disco-interacting protein 2 (Dip2)-GFP control.

measured TAG subspecies levels in the presence of cell wall stress by CFW and observed that there is no depletion in the cognate TAG species suggesting that selective DAG accumulation upon CW stress is not due to TAG lipolysis (Fig. 2—fig. supplement 1B).

Due to the seemingly high selectivity of DAGs for Pkc1 signalling activation in vivo, we sought to answer whether this specificity for C36:0 and C36:1 DAGs is engraved in the DAG binding domain of Pkc1. Therefore, to unravel the specificity of Pkc1 to interact with DAG across subspecies level, we performed an in vitro lipid binding assay. For this, we purified the DAG-binding domain of yeast Pkc1, namely conserved region domain 1 (C1domain), expressed under the galactose-inducible promoter in yeast, with GFP at its C-terminal (*Figure 2—figure supplement 2A*). Then, we performed a liposome sedimentation assay where we prepared liposomes containing phosphatidylcholine (PC) with DAGs of different acyl chain compositions (DAGs were restricted to commercially available ones that are also abundantly present in yeast. C36:1 DAG is commercially unavailable). After incubating different liposomes with the protein, we separated both lipid-bound and unbound fractions and probed with western blot (*Figure 2B*). Surprisingly, we observed binding only in the fraction containing C36:0 liposomes, while there was no binding seen in any other DAG-containing liposome (*Figure 2C*). Furthermore, we also observed a concentration-dependent increase in the binding of C1 domain with C36:0 DAGs (*Figure 2D and F*), confirming their direct physical interaction. On the other hand, no effect on binding was seen even at the higher concentrations of the other DAG species containing liposomes (*Figure 2E–F*). This points to a clear specificity of the C1 domain of yeast Pkc1 for the selective DAG subspecies acted upon by Dip2. A negative control for GFP was also used to rule out any possibility of non-specific binding with GFP (*Figure 2—figure supplement 2B*).

We further asked whether a similar kind of selectivity of Pkc1 C1 domain for Dip2-regulated DAGs exists in higher eukaryotes as well. We cloned the C1 domain of one of the novel Pkc1s of *Drosophila melanogaster* (PKC98E) and rat (PKCδ), as yeast Pkc1 knockout phenotype has been shown to be rescued by mammalian novel PKCs (PKCδ) (*Saiz-Baggetto et al., 2023*), suggesting functional

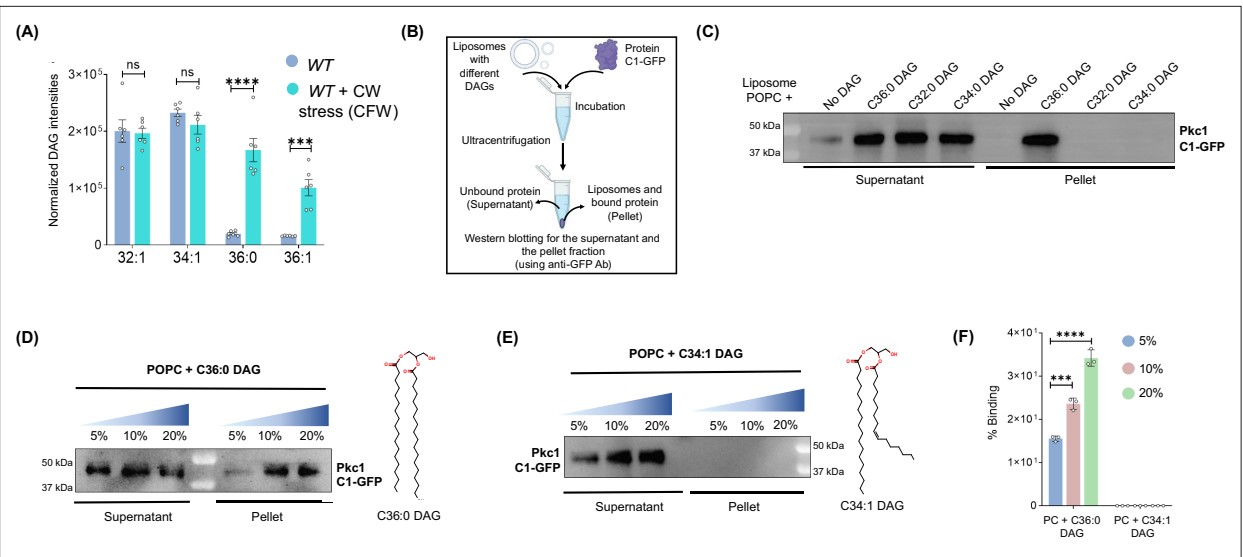

**Figure 2.** Disco-interacting protein 2 (Dip2)-regulated selective diacylglycerol (DAG) subspecies are associated with the activation of protein kinase C (PKC) signalling. (**A**) Lipidomic analysis of wild-type (*WT*) treated with calcofluor white (CFW) (50 µg/mL) for 30 min. DAG intensities normalised to the total protein content in *WT* + CFW have been compared to the intensities from the same species in *WT* control sample. Data are represented as mean ± standard error of mean (SEM) (unpaired, two-tailed Student's t-test; N=6). ****p<0.0001; ***p<0.001 ns=not significant. (**B**) A flowchart for the liposome sedimentation assay explaining the incubation of liposomes with protein, followed by ultracentrifugation and western blotting for supernatant (unbound protein) and pellet (bound protein) fraction. (**C**) Representative western blot for liposome sedimentation assay with different DAGs showing supernatant fraction (unbound protein) and pellet fraction (liposome bound protein), probed for C1 domain using anti-GFP antibody (43 k Da). N>3. (**D**) Western blot probing for C1 domain bound to different concentrations of C36:0 DAG containing liposomes (pellet fraction) and the unbound protein (supernatant fraction) using anti GFP antibody. Structure of respective DAGs is shown on the right side. N>3. (**E**) Western blot probing for C1 domain at different concentrations of C34:1 DAG containing liposomes in both pellet and supernatant fractions. Structure of respective DAGs has been shown on the right side. N>3. (**F**) Bar graph showing quantification of percentage binding of C1-GFP with increasing concentrations of specific (C36:0) and non-specific (C34:1) DAGs. Data are represented as mean ± SD (unpaired, two-tailed Student's t-test; n=3; ***p<0.001; ns = not significant), N=3.

The online version of this article includes the following source data and figure supplement(s) for figure 2:

**Source data 1.** Quantification of diacylglycerols (DAGs) in wild-type (WT) and calcofluor white (CFW)-treated WT.

**Source data 2.** Western blot for C1-GFP incubated with different diacylglycerols (DAGs) containing liposomes.

**Source data 3.** Western blot for C1-GFP incubated with different diacylglycerols (DAGs) containing liposomes.

**Figure supplement 1.** pSlt2 levels and TAG quantification in WT under cell wall stress condition.

**Figure supplement 1—source data 1.** PDF file containing original western blots for *Figure 2—figure supplement 1A*, indicating the relevant bands.

**Figure supplement 1—source data 2.** Original files for western blot analysis displayed in *Figure 2—figure supplement 1A*.

**Figure supplement 1—source data 3.** Excel file containing raw values for triacylglycerol (TAG) intensities displayed in *Figure 2—figure supplement 1B*.

**Figure supplement 2.** PKC C1 GFP expression and Negative control for GFP in liposome sedimentation assay.

**Figure supplement 3.** Lipisome sedimentation assay for PKC C1 domains form Drosophila melanogaster (C1 98E-GFP) and Rattus novegicus (C1δ-GFP).

**Figure supplement 3—source data 1.** PDF file containing original western blots for *Figure 2—figure supplement 3A-C* indicating the relevant bands.

**Figure supplement 3—source data 2.** Original files for western blot analysis displayed in *Figure 2—figure supplement 3A-C*.

homology between the two. After expressing and purifying the C1 domains (*Figure 2—figure supplement 3A*), we performed the liposome sedimentation assay and observed that PKC98E C1 domain binds the DAGs which were observed to be majorly accumulated in ΔDmdip2 i.e., C32:0 and C34:1 DAGs (*Mondal et al., 2022*). Interestingly, C36:0, which was not changed significantly on deleting *DIP2* in *Drosophila*, is also found to bind the C1 domain of PKC 98E (*Figure 2—figure supplement 3B*). This could possibly be due to the difference in the in vivo and the in vitro DAG specificity of Dip2 and Pkc, respectively. Since the lipidomic profile of the *Drosophila DIP2* deleted strain depicts the DAG changes from its whole-body tissue, it fails to provide an accurate picture of tissue-specific DAG specificity of Dip2, which would correspond to PKC.

Similarly, for rat PKCδ C1 domain, the liposome binding assay resulted in its binding with all the three DAGs tested (C32:0, C34:1, C36:0) (*Figure 2—figure supplement 3C*) which also corroborates our previous finding, where we observed the accumulation of multiple DAGs on knocking-out *DIP2A* from mouse embryonic cells (*Mondal et al., 2022*). Though this indicates a lack of DAG-specificity in Dip2 and PKC of higher eukaryotes, the presence of multiple paralogs of Dip2 and PKC isoforms complicates the identification of clear DAG selectivity. However, the correlation between the lipidomics data from Dip2A knockout mouse embryonic cells and the liposome sedimentation assay for PKCδ further strengthens our hypothesis that Dip2-regulated DAG species are involved in activation of PKCs across life forms.

## The canonical DAG metabolism axis is not involved in Pkc1 signalling

Since selective DAG-metabolism by Dip2 is associated with Pkc1 signalling, we asked whether manipulating the levels of bulk DAGs inside cells, mainly produced through canonical DAG metabolism can regulate Pkc1 signalling. DAG, being a central intermediary molecule in lipid metabolism, is acted upon by several conserved enzymes for phospholipid biosynthesis or storage lipid (TAGs) production. Majority of the enzymes participating in these processes are known to act on the bulk pool of DAGs, while the specificity for acyl-chain length remains to be elucidated (*Li et al., 2020*; *Rockenfeller et al., 2018*). Therefore, we probed two main TAG-forming enzymes, DAG acyltransferase (Dga1) and DAG transacylase (Lro1) along with the key membrane biogenesis enzyme, DAG kinase (Dgk1) for studying the effect of bulk DAG pool on Pkc1 activation and CWI pathway. Our lipidomic analysis of these gene knockouts confirmed that both Dga1 and Lro1 act on almost all the major DAG species and their absence led to ~2–10 fold accumulation of DAGs, consisting of varying chain lengths (*Figure 3A and B*). Δ*dgk1*, on the other hand, did not affect the steady-state DAG levels in yeast (*Figure 3C*), which is in agreement with earlier studies (*Adeyo et al., 2011*; *Li et al., 2020*). Surprisingly, none of the mutants of these bulk DAG-acting enzymes showed cell wall stress resistance in the presence of CFW (*Figure 3D*) or CR (*Figure 3—figure supplement 1*). Quantifying the pSlt2 level in these knockouts also resulted in no change, when compared to total Slt2 levels (*Figure 3E*).

To understand how Dip2 responds to the accumulation of bulk DAGs inside the cells, we probed the localisation of Dip2 in the absence of *LRO1* and *DGA1*. Upon co-staining the cells with Mitotracker Red, we found that Dip2 remains in the mitochondria (*Figure 3—figure supplement 2*). This suggests that the bulk DAG accumulation does not affect Dip2 subcellular localisation. This might also be due to the spatial segregation of the bulk DAG pool in the endoplasmic reticulum from the selective DAG pool, which is likely to be present in mitochondria or mitochondria-vacuole contact sites.

In addition, we also generated a double deletion mutant of *DGA1* and *LRO1* and quantified the DAG levels (*Figure 3F*). Spot assay for the double knockout shows that they are sensitive to cell wall stress in the presence of CFW (*Figure 3G*) and in the presence of CR (*Figure 3—figure supplement 2B*). However, there was ~1.6 fold increase in pSlt2 level compared to *WT*, which is half of what we observe in Δ*dip2* (threefold) (*Figure 3H*). Thus, despite having a ~2 to ~3.5 fold accumulation of total DAG pool in the absence of canonical enzymes, Pkc1-mediated CWI signalling remained unaffected. These observations establish that there is a functional dichotomy in the cellular DAG pools: a metabolically active DAG pool that only participates in lipid biogenesis activities and signalling DAG pool that activates Pkc1 signalling cascade. Therefore, it can be proposed that the Dip2-mediated selective DAG axis has evolved to utilise a DAG subpopulation as a secondary messenger to facilitate the regulation of PKC activation.

## Dip2-regulated DAG subspecies for Pkc1 activation originate from phosphatidylinositol and its acyl-chain remodelling by Psi1

Since canonical DAG metabolism pathway is decoupled from Pkc1 signalling, we sought to identify the source of Dip2-regulated DAG subspecies involved in Pkc1 activation. We probed major metabolic routes that can possibly contribute to specific DAG accumulation in Δ*dip2*. We treated *WT* and Δ*dip2* with various inhibitors with the aim to cut off the supply of DAGs from respective lipids (shown as schematic in *Figure 4A*) and checked for the reduction in DAG accumulation. Interestingly, inhibition of PI(4,5)P$_2$ (hereafter referred to as PIP2) to DAG conversion by U73122, a Phospholipase C (PLC) inhibitor (*Banfic et al., 2013*; *Jun et al., 2004*), reduced the specific DAG levels by 80–90% in a concentration-dependent manner (*Figure 4B*), while the other DAG species remained unaffected

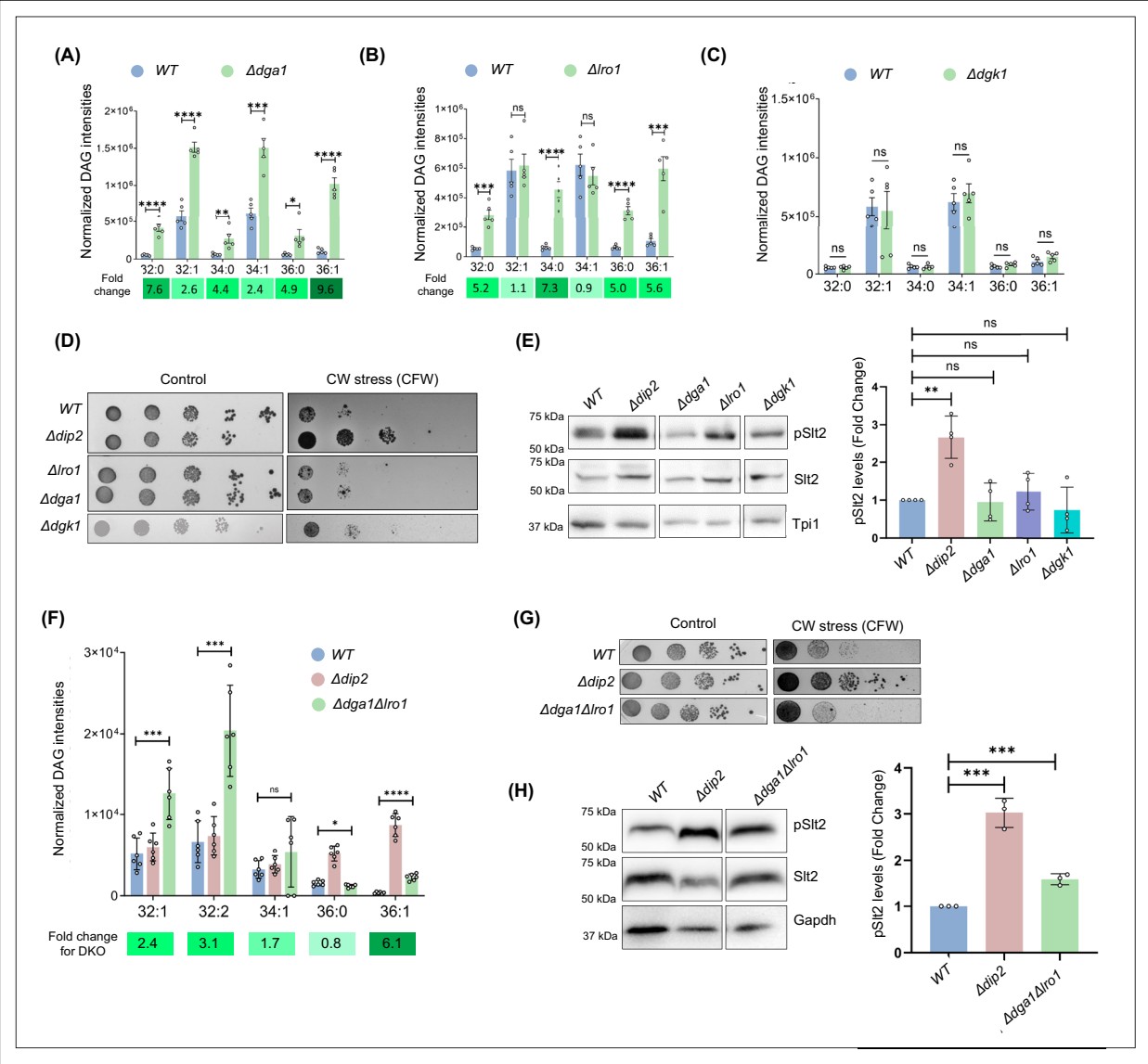

**Figure 3.** Protein kinase C (PKC) activation is independent of the bulk diacylglycerol (DAG) metabolism pathway. (**A–C**) Lipidomic analysis of DAGs in *Δdga1*, *Δlro1*, and *Δdgk1*, compared to wild-type (*W*)*T*. Fold change values are mentioned below and are represented as green colour gradient. Data are represented as mean ± SEM (unpaired, two-tailed Student's t-test; N=5) ****p<0.0001; ***p<0.001; **p<0.01; *p<0.05; ns = not significant. (**D**) Serial dilution assay for single knockouts of bulk DAG acting enzymes (Dga1, Lro1 and Dgk1) in the presence of calcofluor white (CFW) (50 µg/mL), compared to synthetic complete (SC) media control plate. N=3. (**E**) Representative western blot and quantification of pSlt2 (56 kDa) levels in deletion strains of DAG metabolising enzymes. Fold change has been quantified with respect to the total Slt2. Triose phosphate isomerase (Tpi1) (27 kDa) has been used as a loading control. Data are represented as mean ± SD (unpaired, two-tailed Student's t-test); N=4; ****p<0.0001; *p<0.05; ns = not significant. (**F**) DAG subspecies quantification using lipidomics for the double knockout of *LRO1* and *DGA1*, compared to *WT* and *Δdip2*. Data are represented as mean ± SEM (unpaired, two-tailed Student's t-test); n=6; ****p<0.0001; ***p<0.00; *p<0.05; ns = not significant. (**G**) Serial dilution assay image for the cell wall stress sensitivity of *Δdga1Δlro1*, compared to *WT* and *Δdip2*, in the presence of CFW (50 µg/mL), compared to SC media control plate. N=3. SC media control image is reused in *Figure 3—figure supplement 2*. (**H**) Representative western blot for pSlt2 (56 kDa) estimation in *Δdga1Δlro1*, compared to total Slt2 and a loading control Glyceraldehyde-3-phosphate dehydrogenase (Gapdh) (36 kDa). Fold change has been quantified with respect to the total Slt2. Data are represented as mean ± SD (unpaired, two-tailed Student's t-test); N=3; **p<0.01; ns = not significant.

The online version of this article includes the following source data and figure supplement(s) for figure 3:

**Source data 1.** Quantification of diacylglycerol (DAG) in *Δdga1*, *Δlro1*, *Δdgk1*, and *Δlro1Δdgk1 DKO*.

**Source data 2.** Spot assay for single knock outs of diacylglycerol (DAG) acting enzymes.

**Source data 3.** Spot assay for single knock outs of diacylglycerol (DAG) acting enzymes.

**Source data 4.** Western blots and quantification for single knockouts of diacylglycerol (DAG) acting enzymes.

*Figure 3 continued on next page*

*Figure 3 continued*

**Source data 5.** Western blots and quantification for single knockouts of diacylglycerol (DAG) acting enzymes.

**Source data 6.** Spot assay for double knockouts of *LRO1* and *DGA1 in* the presence of calcofluor white (CFW).

**Source data 7.** Spot assay for double knockouts of *LRO1* and *DGA1 in* the presence of calcofluor white (CFW).

**Source data 8.** Western blots and quantification of pSlt2 for double knockouts of *LRO1* and *DGA1*.

**Source data 9.** Western blots and quantification for single knockouts of diacylglycerol (DAG) acting enzymes.

**Figure supplement 1.** Serial dilution assay for DGA1, LRO1 and DGK1 deletions under cell wall stress condition.

**Figure supplement 1—source data 1.** PDF file containing original spot assay plate images for *Figure 3—figure supplement 1A*, indicating the relevant spots and treatments.

**Figure supplement 1—source data 2.** Original files for spot assay plate images displayed in *Figure 3—figure supplement 1A*.

**Figure supplement 1—source data 3.** PDF file containing original spot assay plate images for *Figure 3—figure supplement 1B*, indicating the relevant spots and treatments.

**Figure supplement 1—source data 4.** Original files for spot assay plate images displayed in *Figure 3—figure supplement 1B*.

**Figure supplement 2.** Live cell microscopy images showing the DIC images (left), disco-interacting protein 2 (Dip2) tagged with GFP (green), Mitotracker Red (red), and merged images for red and green channels (right) for (**A**) wild-type (*WT*), (**C**) *Δlro1*, and (**E**) *Δdga1*.

(*Figure 4—figure supplement 1A*). On the other hand, blocking PA to DAG as well as ceramide to DAG synthesis using propranolol and aureobasidin-A, respectively (*Breslow et al., 2010*; *Morlock et al., 1991*; *Sasser et al., 2012*; *Starr et al., 2016*), did not show change in DAG levels (*Figure 4—figure supplement 1B–C*). To confirm the same, we also generated double knockout of *DIP2* and the major PA to DAG-producing enzyme genes in the cell, i.e., *PAH1,* and checked for DAG accumulation. We found that, unlike what we observed in propranolol-treated *Δdip2* which showed no change in DAG level, double knockout of *PAH1* and *DIP2* showed ~1.5-to-2-fold increase in all the bulk DAGs (C32:1, C32:2, C34:1) (*Figure 4—figure supplement 2A*). However, we also observed a significant decrease of ~42% in C36:0 and ~81% in C36:1 DAGs. We also probed the effect of *PAH1* deletion on CW stress pathway and Pkc1 activation. We found that *Δpah1* and *Δpah1Δdip2*, both are defective in growth and sensitive to CW stress (*Figure 4—figure supplement 2B*). This phenotype does not correlate with pSlt2 level (*Figure 4—figure supplement 2C*), suggesting that the CW stress sensitivity is not linked to Pkc1 activation and may indicate a pleiotropic defect due to alteration in lipid metabolism. However, given the reduction in the level of selective DAGs in *Δpah1Δdip2*, we cannot rule out the possibility that Pah1 could also be contributing to the selective DAG accumulation in *Δdip2*. Given this observation, we also asked whether the absence of Pah1 could affect subcellular localisation of Dip2. Therefore, we used microscopy to trace Dip2 in *Δpah1* and found no change in its localisation suggesting that Pah1 deletion has no impact on Dip2 localisation (*Figure 4—figure supplement 2D–E*).

We also deleted *PLC1 in Δdip2* and observed a ~75% reduction in C36:0, C36:1 DAGs (*Figure 4C*), with no change in other DAG species**,** suggesting that Plc1-dependent hydrolysis of C36:0 and C36:1 PIP2 pools serves as the predominant source of Dip2-metabolised selective DAGs. Whether the same source of DAGs is responsible for Pkc1 activation was investigated by checking *pSlt2* levels. *Δplc1Δdip2* and U73122-treated *Δdip2* showed depleted levels of *pSlt2*, when compared to *Δdip2*, while Slt2 phosphorylation remained unaffected in *Δplc1* (*Figure 4D*). This suggested that the DAGs regulated by Dip2 are sourced via hydrolysis of corresponding PIP2 and are further involved in activating Pkc1 pathway. However, a further depletion of pSlt2 level in *Δplc1Δdip2*, compared to *Δplc1* was surprising as deletion of *DIP2* should not alter the level of pSlt2 level. This might have resulted from a rewiring of pathways upon deleting both the genes, which possibly led to a pleiotropic effect. These observations were also strengthened by the live-cell imaging of Plc1 that showed that Plc1 partially localises to the mitochondria (*Figure 4—figure supplement 3A–B*) and vacuoles (*Figure 4—figure supplement 3C–D*), similar to Dip2 suggesting that it might be producing the selective DAGs locally for Dip2 to act upon.

Next, we asked how exactly the specific PIP2 pool is maintained inside the cell to source these DAGs. Strikingly, a previous study has reported that the absence of an acyltransferase, phosphatidylinositol stearoyl incorporating 1 (Psi1), leads to severe depletion of C36:0 and C36:1 PIP2 species. Psi1 remodels the neo-synthesised phosphatidylinositol by adding stearic acid (C18:0) at the *sn-1* position

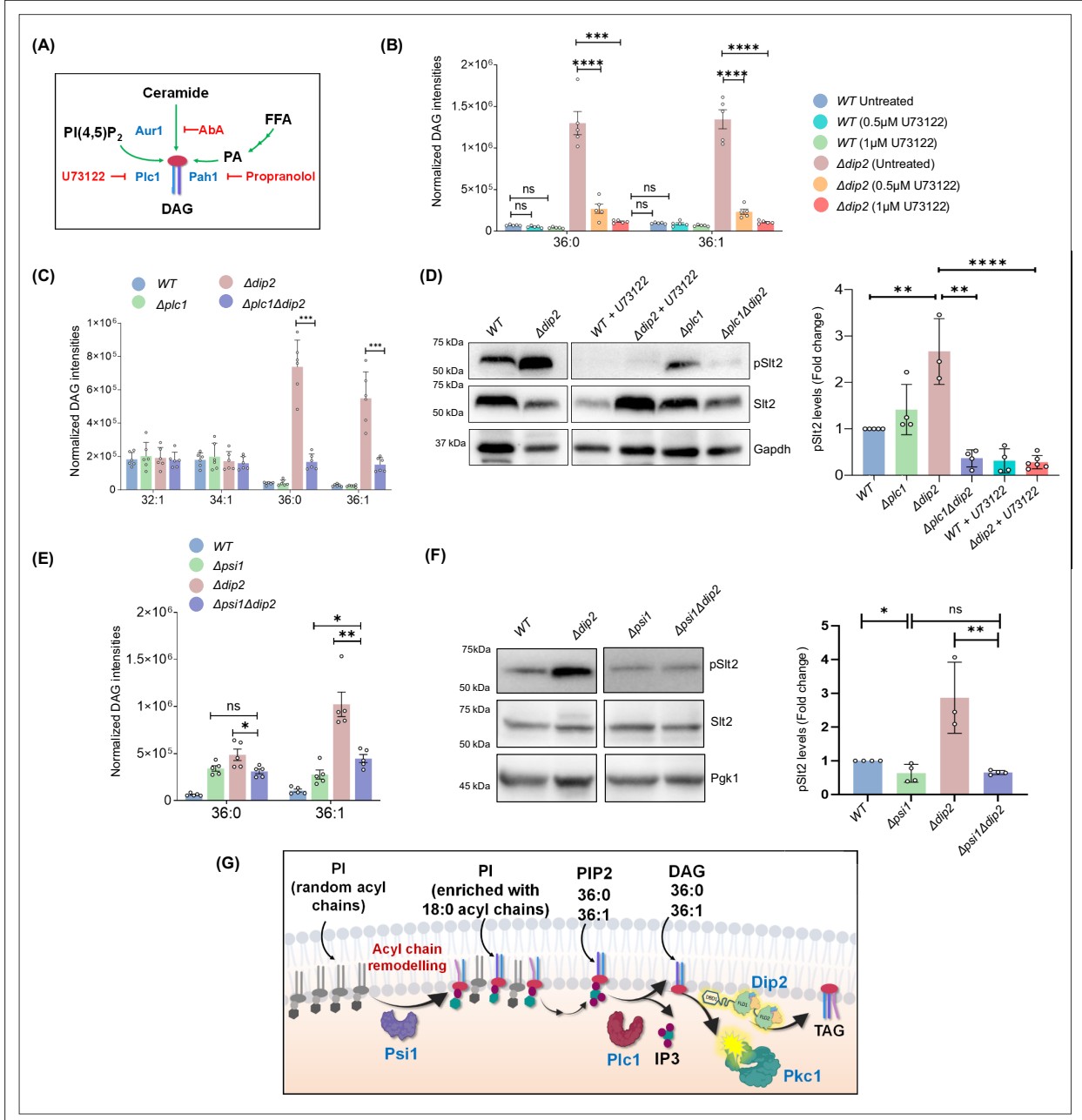

**Figure 4.** Psi1-Plc1-Dip2 triad axis regulates selective diacylglycerol (DAG) subspecies to modulate protein kinase C (PKC) signalling. (**A**) Schematic showing various pathways (green arrows) that feed into the production of DAGs in the presence of various enzymes (shown in blue) and different chemical inhibitors (red) blocking the respective pathways. (**B**) Quantification of selective DAGs in wild-type (*WT*) and *Δdip2* samples treated with two different concentrations, i.e., 0.5 and 1 µM of U73122. Data are represented as mean ± SEM (unpaired, two-tailed Student's t-test; N=5). ****p<0.0001; ***p<0.001; ns = not significant. (**C**) Specific DAG quantification using lipidomics for deletion of *Δplc1* and *Δplc1Δdip2*. Data are represented as mean ± SEM (unpaired, two-tailed Student's t-test; n=6). ****p<0.0001; ***p<0.001 ns=not significant. N=6. (**D**) Representative western blot image and quantification of pSlt2 (56 kDa) levels for indicated samples. Fold change has been quantified with respect to the total Slt2. Gapdh (36 kDa) has been used as a loading control. Data are represented as mean ± SD unpaired, two-tailed Student's t-test; N>4; ****p<0.0001; **p<0.01; ns = not significant. (**E**) Lipidomic analysis of specific DAGs in double knockout of *PSI1* and *DIP2*, compared with *WT* and *Δpsi1*. Data are represented as mean ± SEM (unpaired, two-tailed Student's t-test; **p<0.01; *p<0.05; ns = not significant. N=5). (**F**) Representative western blot image and quantification of pSlt2 (56 kDa) levels for indicated samples. Two data points for controls (*WT* and *Δdip2*) are same as that of **D**. Fold change has been quantified with respect to the total Slt2. Phosphoglycerate kinase (Pgk1) (45 kDa) has been used as a loading control. Data are represented as mean ± SD unpaired, two-tailed Student's t-test; N>3; ****p<0.0001; ***p<0.001; ns = not significant. (**G**) Schematic showing Psi1-Plc1-Dip2 axis to regulate Pkc1 signalling where Psi1

*Figure 4 continued on next page*

*Figure 4 continued*

remodels phosphoinositides (PI) to enrich it with C36:0 and C36:1 containing acyl chain, which is channelled to the selective DAGs via Plc1 and in turn activate Pkc1.

The online version of this article includes the following source data and figure supplement(s) for figure 4:

**Source data 1.** Quantification of diacylglycerol (DAG) in wild-type (WT) and *Δdip2* in the presence of Plc1 inhibitor.

**Source data 2.** Quantification of DAG in double knockout of disco-interacting protein 2 (*DIP2*) and *PLC1*.

**Source data 3.** Western blot and quantification of pSlt2 in the double knockout of disco-interacting protein 2 (*DIP2*) and *PLC1* and in the presence of Plc1 inhibitor.

**Source data 4.** Western blot and quantification of pSlt2 in the double knockout of disco-interacting protein 2 (*DIP2*) and *PLC1* and in the presence of Plc1 inhibitor.

**Source data 5.** Quantification of diacylglycerol (DAG) in double knockout of disco-interacting protein 2 (*DIP2*) and *PSI1*.

**Source data 6.** Western blot and quantification of pSlt2 in the double knockout of disco-interacting protein 2 (*DIP2*) and *PSI1*.

**Source data 7.** Western blot and quantification of pSlt2 in the double knockout of disco-interacting protein 2 (*DIP2*) and *PSI1*.

**Figure supplement 1.** DAG quantification for WT and DIP2 KO treated with various inhibitors.

**Figure supplement 1—source data 1.** Excel file containing raw values for diacylglycerol (DAG) intensities displayed in *Figure 4—figure supplement 1*.

**Figure supplement 2.** DAG quantification and pSlt2 levels in double knockouts of PAH1 and DIP2.

**Figure supplement 2—source data 1.** Excel file containing raw values for diacylglycerol (DAG) intensities displayed in *Figure 4—figure supplement 2A*.

**Figure supplement 2—source data 2.** Original files for spot assay plate images displayed in *Figure 4—figure supplement 2B* with relevant labelling.

**Figure supplement 2—source data 3.** PDF file containing original western blots for *Figure 4—figure supplement 2C* indicating the relevant bands.

**Figure supplement 2—source data 4.** Original files for western blot analysis displayed in *Figure 4—figure supplement 2C*.

**Figure supplement 2—source data 5.** Original microscopy image for *Figure 4—figure supplement 2D*.

**Figure supplement 3.** Partial localization of Plc1 with mitochondria and vacuole.

**Figure supplement 3—source data 1.** Original microscopy image for Plc1-GFP in *Figure 4—figure supplement 3A and C*.

**Figure supplement 4.** DAG subspecies quantification for *Δpsi1* measured via lipidomics.

---

of its glycerol backbone (*Le Guédard et al., 2009*), thus enriching the PI and eventually PIP2 pool with specific acyl chains (*Doignon et al., 2016*). Hence, we checked the DAG levels in *Δdip2* upon deleting *PSI1* (*Δpsi1Δdip2*) and observed a significant decrease in the accumulated DAGs (*Figure 4E*). It is worth noting here that *PSI1* deletion in *Δdip2* background did not lead to complete depletion of accumulated DAGs in *Δdip2*, but a considerable quantity of DAGs, i.e., ~36% of C36:0 and 40% of C36:1, respectively are channelled from Psi1. In the single deletion of *PSI1*, unexpectedly, we observed an increase in almost all the DAGs quantified (*Figure 4—figure supplement 4*), which could be because of a compensation effect the system might have adapted due to a decrease in C36:0, C36:1 phosphoinositide and respective DAGs, as reported earlier (*Doignon et al., 2016*). To test its role in Pkc1 activation, we measured pSlt2 levels in *Δpsi1Δdip2* and found that pSlt2 levels restored back to *WT* levels (*Figure 4F*). In fact, deletion of *PSI1* itself decreases the pSlt2 level in *WT* by ~35%, which phenocopies *Δpsi1Δdip2*, thereby strengthening the contribution of Psi1 to Pkc1 activation.

Together, these observations emphasise that the acyl chain remodelling of PI by Psi1 for specifically enriching C36:0 and C36:1 ensures channelling of these chain length compositions towards respective PIP2. The corresponding PIP2 species are then hydrolysed to selective DAGs by Plc1, which is responsible for Pkc1 activation. Dip2, which acts on these DAGs, regulates Pkc1 signalling and keeps a check by converting these DAGs to TAGs (*Figure 4G*). In addition, these observations also reiterate that the bulk DAG pool cannot activate Pkc1 signalling cascade. Therefore, the Psi1-Plc1-Dip2 axis serves not only as a source of DAGs but also acts as a critical checkpoint for the Pkc1 signalling cascade.

## Co-emergence and coevolution of Dip2-PKC axis underscores the selective nature of DAG-based signalling

DAGs are universally recognised by a structurally conserved domain called C1 domain (conserved domain 1) (*Hurley et al., 1997*), found in different DAG effector proteins including PKC. To understand the evolutionary relationship between DAG effectors and selective DAG regulator, Dip2, we

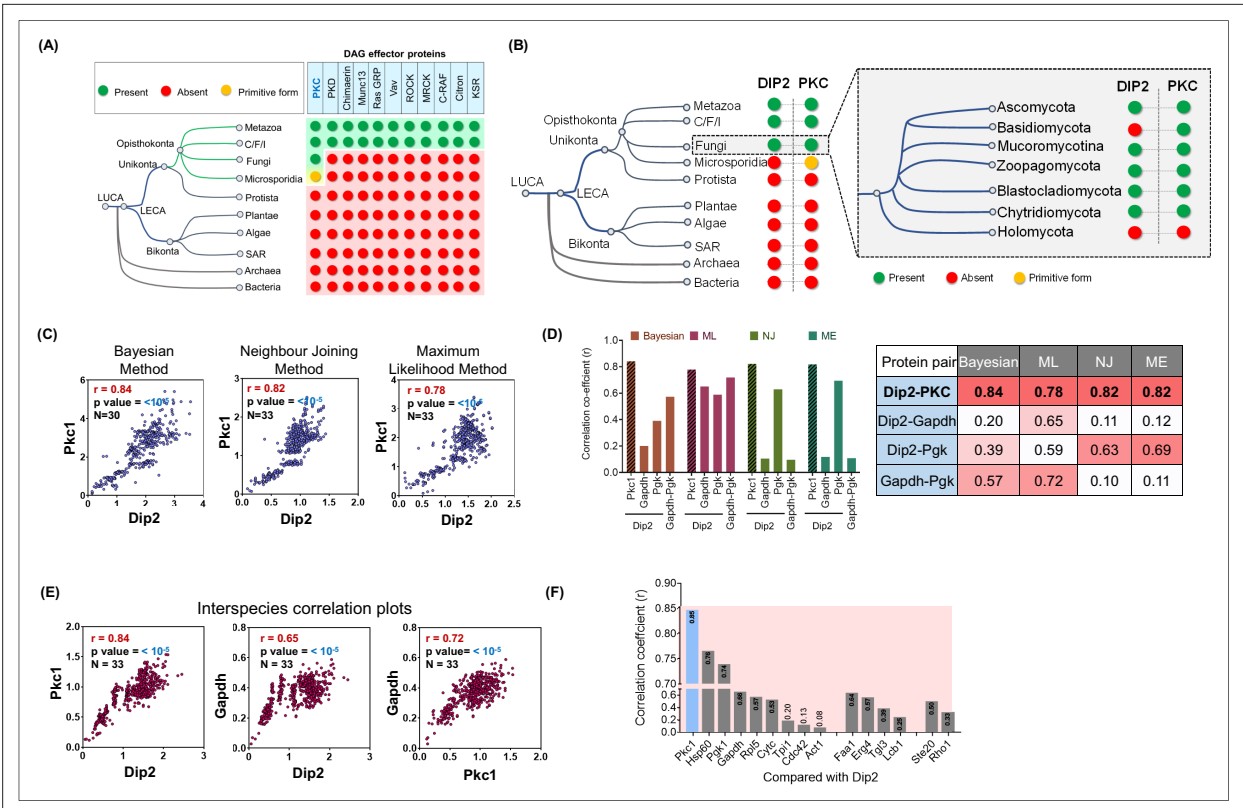

**Figure 5.** Disco-interacting protein 2 (Dip2) and protein kinase C (PKC) share a parallel history of co-emergence coevolution across Opisthokonta. (**A**) Emergence and distribution of diacylglycerol (DAG) effector proteins across tree of life. Green circles represent presence, while red circles represent absence of effector proteins. Yellow circle represents the primitive form of PKC. List of DAG effectors is taken from a previous study (***Colón-González and Kazanietz, 2006***). (**B**) Phylogenetic profiling of Dip2 and PKC shows their co-emergence in Opisthokonta. Inset shows presence of Dip2 and PKC in different fungal branches (***Ocaña-Pallarès et al., 2022***). Green circle represents presence, red represents absence, while yellow circle represents a primitive form of PKC harbouring only C1 and kinase domain. (**C**) Interspecies coevolution plot (using patristic distance) between fungal Dip2 and PKC using three different phylogenetic algorithms, with the correlation coefficient value represented by r. Number of organisms and significance levels are indicated. (**D**) Interspecies correlation coefficient values (r) for Dip2 compared with PKC and control proteins Gapdh and Pgk using various algorithms for calculating phylogenetic distances. r value is calculated for Gapdh and Pgk the same way. The table contains the exact correlation coefficient values for all the above-mentioned protein pairs using different algorithms. (**E**) Coevolution analysis (using genetic distance) between Dip2 and PKC along with their individual comparison with control protein Gapdh. Number of organisms and significance levels are indicated. (**F**) Correlation coefficient values represented as bar graph for Dip2 compared with a set of housekeeping genes and lipid metabolising proteins.

The online version of this article includes the following source data and figure supplement(s) for figure 5:

**Source data 1.** Quantification of interspecies correlation coefficient values for disco-interacting protein 2 (Dip2) and protein kinase C (PKC) using different algorithms.

**Source data 2.** Quantification of interspecies correlation coefficient values for disco-interacting protein 2 (Dip2), protein kinase C (PKC), Gapdh, and Pgk1 using different algorithms.

**Source data 3.** Quantification of interspecies correlation coefficient values using genetic distances for disco-interacting protein 2 Dip2, protein kinase C (PKC), and Gapdh.

**Source data 4.** Quantification of interspecies correlation coefficient values for various housekeeping proteins and disco-interacting protein 2 (Dip2).

**Figure supplement 1.** Phylogenetic analysis and distribution of PKC, PLC and PSI.

**Figure supplement 2.** Coevolutionary analysis of Dip2 and PKC along with control proteins.

**Figure supplement 2—source data 1.** Excel file containing correlation coefficient values for ***Figure 5A–D***.

probed their distribution across the tree of life. Previously, we showed that Dip2 had emerged in early opisthokonts. Interestingly, phylogenetic distribution of all known DAG effectors (***Colón-González and Kazanietz, 2006***) revealed that the majority of these are concentrated in metazoans and metazoan sister groups (Choanoflagellates, Filasterea, and Ichthyosporea), while PKC is the only DAG effector conserved in fungal groups and all opisthokonts (***Figure 5A***).

Fungal Pkc1 (hereafter referred to as prototypical PKC) (*Schmitz and Heinisch, 2003*) harbours all the typical regulatory domains present in higher eukaryotes, which includes HR1 (homology region 1) domain, C2 domain (conserved domain 2), C1 domain (conserved domain 1) along with the protein kinase domain (*Schmitz and Heinisch, 2003*). The phylogenetic distribution suggests that the prototypical PKC had probably undergone multiple gene duplication, domain shuffling, and gene loss events during early metazoan evolution resulting in a repertoire of PKC isoforms (*Figure 5—figure supplement 1A*). Furthermore, we observed the presence of primitive PKC, consisting of only C1 and protein kinase domains in Microsporidia, the sister group of Fungi (*Bass et al., 2018*; *James et al., 2013*). Interestingly, only two *Dictyostelium* species of protists out of ~50 genomes (available in KEGG database) harbour PKC in its primitive form (*Goldberg et al., 2006*; *Keenan et al., 1997*; *Wang et al., 1996*). Thus, the early DAG effectors with C1 domain had evolved in one of the two major eukaryotic branches, known as Unikonta (comprising protist, microsporidia, fungi and metazoa). To our surprise, the other eukaryotic branch, Bikonta, comprising plants, algae, and SAR (**S**tramenopiles, **A**lveolates, and **R**hizaria) organisms, is completely devoid of the DAG effectors (*Figure 5A*). Overall, our analysis suggests that PKC is the first DAG effector to be recruited in eukaryotes and evolved exclusively in opisthokonts.

The emergence of a prototypical Pkc with DAG-binding C1 domain in early opisthokonts suggests a potential requirement of DAG regulators for optimal signalling process. Previously, we have reported that FAAL-like domains from bacteria gave rise to two tandem FAAL-like domain containing protein, Dip2 across fungi and metazoans (*Mondal et al., 2022*; *Patil et al., 2021*). Interestingly, the phylogenetic profile analysis of PKC (*Figure 5—figure supplement 1B*) and Dip2 showed that both are distributed across Fungi and Animalia, but are totally absent from Archaea, Bacteria, Plantae, and Algae, suggesting that both the proteins show evolutionary co-emergence and co-occurrence (*Figure 5B*). It is important to note here that Dip2 is absent in a major phylum of fungi called Basidiomycota, suggesting a possible gene-loss event in this phylogenetic branch. Although Basidiomycota harbours C1 domain containing Pkc1, whether the Pkc1 is regulated by an unknown functional orthologue of Dip2, or a distinct Pkc1 signalling has been evolved in this branch of fungi, is yet to be understood. Taken together, our analysis underscores the fact that the emergence of a fully functional PKC signalling pathway can be marked by the presence of Dip2 from fungi onwards.

To investigate the evolutionary history of Psi1-Plc1 axis, we examined the phylogenetic distribution pattern and found that Psi1 was evolved in early eukaryotes and is conserved in both Unikonta and Bikonta. On the other hand, Plc1, consisting of multiple domains such as PH domain, EF hand domain, PI-PLC domain, and C2 domain, emerged in protists and is distributed across Unikonta, while absent in Bikonta (*Figure 5—figure supplement 1C*). This evolutionary history parallels that of the relevant DAG effector, PKC.

We then sought to quantitatively assess the phenomenon of co-evolution between Dip2 and PKC. To investigate this, we furthered our bioinformatic analysis using molecular phylogenies of Dip2 and PKC from the representative fungi across fungal divisions (*Ocaña-Pallarès et al., 2022*). Correlation between the phylogenetic distances (patristic distance) of Dip2 and PKC were calculated using multiple methods like the Bayesian method, neighbour joining, minimum-evolution, and maximum likelihood (*Fourment and Gibbs, 2006*). The result showed significantly high association in the pairwise patristic distances of Dip2 and PKC, compared to other control gene pairs like Pgk1 and Gapdh (*Figure 5C–D*), suggesting their possible co-evolution across fungal phylogenetic branches.

To calculate the interspecies correlation of genetic divergence rate, we adopted the approach described for mirrortree algorithm (*Edgar et al., 2012*; *Ochoa and Pazos, 2014*; *Pazos and Valencia, 2001*). In this case, genetic distances (distance matrix-based) were used to estimate the evolutionary association between Dip2 and PKC (*Figure 5E*), which scored more than a correlation coefficient of 0.8, an empirical cut-off value suggested by *Pazos and Valencia, 2001*. We further reconfirmed the above observation using a larger set of all available fungal genomes (*Figure 5—figure supplement 2A*). This observation is further substantiated by a lesser correlation obtained on comparing Dip2 or PKC with a highly conserved gene like *GAPDH* (*Figure 5E*). Furthermore, the correlation between *DIP2* or *GAPDH* and a diverse set of conserved genes with related and unrelated functions was found to be below the cut-off value of correlation coefficient (<0.8) (*Figure 5*, *Figure 5—figure supplement 2*). We also confirmed that the correlation between Dip2-PKC pair is significantly different from this random control set (*Figure 5—figure supplement 2C–D*). Therefore, a significant interspecies

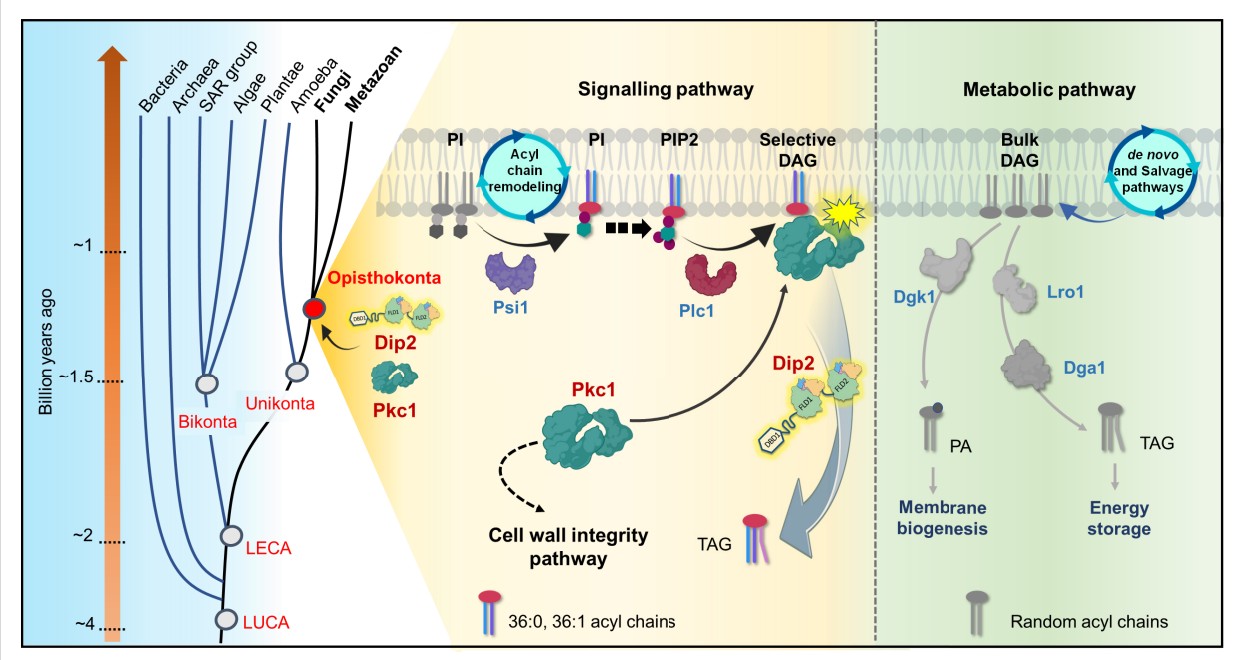

**Figure 6.** A graphical representation depicting the emergence of disco-interacting protein 2 (Dip2)-protein kinase C (PKC) axis and the evolution of selective diacylglycerol (DAG)-based PKC signalling in Opisthokonta. Dip2 is recruited parallel to PKC in the tree of life at the root of Opisthokonta evolution ~1.2 billion years ago. This Dip2-PKC axis remains conserved across Fungi and Metazoan. Two distinct DAG pools are sourced from de novo or salvage pathways and acyl chain remodelling of PIs, leading to metabolic and signalling DAG pools, respectively. Remodelled phosphoinositides (PI, PIP, PIP2) by Psi1 are enriched with 18:0 acyl chains and channelled to corresponding PIP2, which forms respective DAG species (C36:0, C36:1) upon hydrolysis by Plc1. These selective DAG subspecies act as secondary messenger for PKC signalling and, therefore, regulate the CWI pathway in yeast. Dip2 maintains the levels of these selective DAGs by facilitating their conversion to triacylglycerols (TAGs), thereby creating a diversification of DAG function into metabolic and signalling pathways.

correlation between Dip2 and PKC suggested that both the genes had experienced a similar evolutionary pressure, probably due to their participation in the same physiological axis. Overall, the evolutionary analysis suggests that Dip2 and PKC are phylogenetically correlated and have coevolved together for successful establishment of selective DAG-based PKC signalling in Opisthokonta.

## Discussion

The discovery of Dip2 as a conserved and selective DAG regulator presented a cellular scenario, where there are two 'functional' pools of DAG in cell, *viz.*, the bulk 'metabolic' pools of DAG and the minor pools of 'signalling' DAG. The former is involved in membrane biogenesis and storage lipid biosynthesis, while the later is Dip2-regulated selective DAG pool, responsible for activating Pkc1 signalling. Here, we provide evidence for the emergence of Dip2 and the functional diversification of DAG pool, responsible for activation of Pkc1 signalling cascades in yeast. As DAG metabolising enzymes are generally promiscuous, their recruitment for selectively activating and fine-tuning the Pkc1 signalling events is a distant possibility. The rooting of a lipid-based signalling in eukaryotes during early Opisthokonta evolution possibly necessitated a more precise regulatory mechanism, making a species-specific regulator such as Dip2 inevitable for DAG-based PKC signalling (*Figure 6*).

The metazoan PKC isoforms such as conventional and novel PKCs are well characterised to be regulated by signalling molecules such as DAGs, calcium, phosphatidylserine, etc. (*Rosse et al., 2010*). Different combinations of regulatory domains in PKC isoforms result in highly diversified and mutually exclusive functions in animals (*Reyland, 2009*). Interestingly, the single prototypical PKC of yeast with all the regulatory domains is involved in a myriad of physiological processes, for example, cell wall organisation, progression of cell cycle, lipid metabolism, cell polarity, p-body assembly, septin organisation etc. (*Dey et al., 2017*). Initially, Pkc1 activation was shown to be unaffected by DAGs (*Antonsson et al., 1994*), while a few recent reports have suggested the involvement of DAGs in

activating Pkc1 in vitro (*Dey et al., 2017*). We believe the discrepancy among the existing reports has been cleared as we have observed a selective DAG-mediated Pkc1 and CWI pathway activation in yeast. The earlier observed mutations in the putative DAG-binding C1 domain of yeast Pkc1 which result in decreased fitness in cell wall stress conditions (*Jacoby et al., 1997*), corroborates our findings on the crucial role of DAG in activation of Pkc1-based CWI pathway. Similarly, a differential role of other domains like HR1 has been proposed in earlier studies (*Schmitz et al., 2002*). Whether the selective DAG-based Pkc1 activation also requires HR1 domain to bind Rho1, remains to be seen.

Recent reports have shown that exogenous addition of DAGs with varied acyl chains has a differential effect on the translocation of mammalian PKC isoforms (*Schuhmacher et al., 2020*). While this indicates the role of distinct acyl chain-containing DAGs in differentially activating PKC isoforms, it does not represent a physiological scenario where a cell employs this. Here, we have shown a selective increase in C36:0 and C36:1 DAGs which results in activation of Pkc1 upon subjecting the yeast cells to cell wall stress. Additionally, the DAG binding domain of Pkc1 shows binding exclusively with C36:0 DAG and not with other DAGs in vitro. Hence, we have identified the specific DAG subspecies for regulating Pkc1, thereby, bridging the gap in the understanding of selective DAG-driven PKC activation, that was speculated 30 y ago (*Marignani et al., 1996*). Nonetheless, the rationale for selecting the two DAG subspecies out of numerous possible combinations of acyl chains for activating Pkc1 remains an enigma and needs to be probed. It also opens up possibilities for specific roles of selective species of lipids that perhaps explains why there is a preservation of diversity in lipidome during evolution of different life forms.

The canonical DAG metabolising enzymes, despite acting on C36:0, C36:1 DAGs to a certain level, are excluded from their regulatory role in Pkc1 signalling. Although the double deletion of *LRO1* and *DGA1* elevates Pkc1 activation to an extent, this increase is insufficient to confer resistance to cell wall stress. Whether it is the distinct source of Dip2 regulated DAGs from the PI pool or the spatial segregation of the two pools inside the cell, which make it exclusive to Pkc1 pathway regulation needs to be probed. Furthermore, how Pkc1, which is a bud-site (plasma membrane) protein in yeast (*Denis and Cyert, 2005*), is regulated by a selective DAG regulator Dip2 residing at mitochondria-vacuole contact site (*Mondal et al., 2022*) requires further investigation. Characterising the spatiotemporal nature of specific DAG accumulation that drives these processes would provide clues to the partitioning of Pkc1 activity and is currently underway.

Although, PKC is a well-studied signalling proteins and DAG-PKC signalling has been a textbook model for several decades, the evolutionary and regulatory origin of this pathway remained unknown. Our current work provides compelling evidence that suggests that PKC was the first DAG effector protein to be evolved and conserved in opisthokonts, while completely absent in the other eukaryotic supergroup Bikonta which comprises plant, algae and SAR organisms. The evolution of such distinct lipid-based signalling in early eukaryotic ancestors is probably a decisive factor for further branching into two subgroups- Unikonta (mainly Opisthokonta) and Bikonta. Thus, the work also suggests that the emergence of Dip2 around 1.2 billion years ago in Opisthokonta was a unique metabolic innovation to establish a precisely regulated PKC signalling pathway (*Figure 6*).

The role of Dip2 in fungal virulence has been highlighted in several studies as its deletion in various plant and animal pathogenic fungi like *Magnaporthe oryzae* (causing rice blast), *Cochliobolus heterostrophus* (causing southern corn blight) and *Coccidioides posadasii* (causing valley fever) renders them avirulent (*Lu et al., 2003*; *Narra et al., 2016*; *Wang et al., 2016*). As PKC signalling has also been shown to be crucial for pathogenicity of fungi, the identification of Dip2 as a unique regulator of PKC would open avenues for therapeutic and pharmacological interventions. Similarly, the phenotypes for PKC hyperactivation and deletion of Dip2 observed independently in animals suggest a link between Dip2 and regulation of PKC isoforms. For instance, the abnormalities observed in upregulation of different PKC isoforms such as axonal bifurcation and outgrowth (*Nitta et al., 2017*; *Zhang et al., 2019*), altered dendritic spine morphogenesis (*Calabrese and Halpain, 2005*; *Ma et al., 2019*) etc. phenocopy the defects reported in Dip2 knockouts (*Ma et al., 2019*; *Nitta et al., 2017*). Moreover, both Dip2A and PKC have been implicated independently in autistic behaviour in model organisms (*Liu et al., 2018*; *Ma et al., 2019*; *Philippi et al., 2005*). Similarly, loss of Dip2C, a candidate breast cancer gene (*Li et al., 2017*), leading to increased epithelial-mesenchymal transition (EMT) in cancer cell lines (*Larsson et al., 2017*), also resembles PKCε overexpression, which not only induces EMT but is also a biomarker for aggressive breast cancer (*Jain and Basu, 2014*). These observations provide

a rationale to test the possibility that multiple paralogs of Dip2 might be regulating various PKC isoforms by acting on specific pool of DAGs in higher eukaryotes.

## Materials and methods

### Yeast strains and plasmids

Yeast strains used in this study are all BY4741 (MATa ura3Δ0 his3Δ1 leu2Δ0 met15Δ0) (*Brachmann et al., 1998*) derived from S288C genetic background as listed in (*Supplementary file 1*). Single gene knockout strains are retrieved from the public repository Euroscarf, except for *Δdip2* which is generated by homologous recombination from our previous study (*Mondal et al., 2022*).

Galactose promoter was replaced by the native Dip2 promoter in the plasmid pYSM10 used in our previous study using Gibson assembly cloning. This was also used as the template for site-directed mutagenesis by performing polymerase chain reaction using mutagenic oligonucleotides. These mutants were confirmed by sequencing.

Double knockout strains of *Δpsi1Δdip2* and *Δplc1Δdip2* were generated by knocking out *DIP2* from *Δpsi1* and *Δplc1* strains from Euroscarf library by homologous recombination. Briefly, *Dip2* (Cmr2; SGD ID: S000005619) gene was replaced with a PCR amplified hygromycin resistance cassette from pFA6A-hphMX6 plasmid. The primers were designed in such a way that the PCR product will have flanking sequences homologous to the 5′ and 3′ end of *DIP2* gene. The PCR product was purified and transformed using direct carrier DNA/PEG method-based transformations (*Gietz and Schiestl, 2007*). Transformed cells were plated on both hygromycin and G418 selection plate for *DIP2* and *PSI1/PLC1* deletion respectively. Thus, the resistant colonies were further validated for deletion using PCR primers amplifying the hygromycin cassette, designed using a web tool Primers-4-Yeast (*Yofe and Schuldiner, 2014*) as listed in (*Supplementary file 2*). Dip2 was tagged with GFP in the knockout strains of *LRO1* and *PAH1* by transforming amplicons produced by using Dip2-specific C-tag primers, using pFA6A-HisMX6 plasmid. For generating Dip2-GFP in *Δdga1*, we transformed Dip2 tagged with GFP under its native promoter in the double knockout of *DIP2* and *DGA1*. All the GFP tagging was confirmed by western blot using anti-GFP antibody. For C1a-C1b protein expression, C1a-C1b (413–534 aa) was amplified from yeast genomic DNA and cloned into galactose inducible pYSM5 vector used in our previous study (*Mondal et al., 2022*).

### Media and reagents

Yeast cells were grown in synthetic complete (SC) media, at 30 °C in a shaking incubator with 200 rotations per minute (RPM). SC media constituents include 1.7 g/l yeast-nitrogen base with ammonium sulphate (BD Difco), 20 g/l glucose supplemented with appropriate amino acids (Sigma), adenine (Sigma A5665), and uracil (Sigma U0750). G418 (200 µg/mL) and Hygromycin (0.2 mg/mL) were used for knockout strain selection. For some experiments, YP media (yeast extract – 10 g/l, peptone – 20 g/l) supplemented with 2% dextrose was used. 2% bacto agar along with the above-mentioned SC media constituents was used for solid media.

### Spot assay

An overnight primary culture in SC media was diluted to 0.5 $OD_{600}$ nm and grown till the secondary culture reaches $OD_{600}$ nm of ~2.0. This culture was diluted to 0.2 $OD_{600}$ nm and was used as the first dilution, which was further diluted into multiple serial dilution stocks ($10^{-1}$, $10^{-2}$, $10^{-3}$, and $10^{-4}$). 10 µl of each dilution stocks were spotted sequentially on SC agar control plate and SC agar with either Congo red (100 µg/mL) (Sigma) or Calcofluor white (Sigma) (50 µg/mL) to induce cell wall stress or in combination with cercosporamide (2 µg/mL).

### Colony forming unit assay

A primary culture was inoculated overnight and was diluted to 0.2 $OD_{600}$ nm in fresh SC media, cultured till the early log phase or $OD_{600}$ nm of ~0.8. Cells were serially diluted to 0.002 $OD_{600}$ nm in autoclaved filtered Milli-Q water and 80 µL of the cells were plated on SC media agar as well as SC media agar with CR. After incubation for ~48 hr at 30 °C, number of colonies were counted and CFU was calculated.

## Growth curve analysis

The overnight grown primary culture of *WT* and *Δdip2* was inoculated in fresh SC media with ~0.2 $OD_{600}$ nm. Growth curve was performed using this secondary culture grown with and without CR. The cell density was measured starting from an $OD_{600}$ nm of 0.2 at zero time point to 28 hr. The readings were recorded at an interval of every 4 hr, until it reaches the stationary phase.

## Protein expression and purification

C1-GFP (413–534 amino acids) tagged with GFP and 8xHis at its C-terminus was expressed in *S. cerevisiae*. Wherein, 8 L cell culture grown in YP-Galactose media for induction was harvested at 4000 RPM and was resuspended in buffer A (PBS pH 8.0, 500 mM NaCl, 10% Glycerol and 5 mM β-mercaptoethanol), protease inhibitor cocktail (PIC), 1 mM phenylmethysulfonyl fluoride (PMSF) and lysed using 0.5 mm glass beads. For the expression check, cell lysate was mixed with 6X-SDS loading dye and was run on 12% SDS gel and checked for in-gel GFP fluorescence. The protein was purified by affinity chromatography with nickel-nitrilotriacetic acid (Ni-NTA) agarose beads using buffer A. Protein was eluted by single-step elution using Buffer A containing 250 mM imidazole. Following elution, the protein was buffer exchanged in Buffer B (PBS, 150 mM NaCl, 10% Glycerol, 2 mM DTT), concentrated, and stored at –80 °C until further use.

The C198E (a.a. 171–304 of *Drosophila* PKC98E) and C1 delta (a.a. 152–280 of mouse PKC delta) genes were cloned into pETite vector (Lucigen, USA) with a C-terminal TEV cleavage site, GFP and 6 x His tag. *E. coli* HI-control BL21(DE3) (Lucigen, USA) cells were used for expression using Isopropyl β- d-1-thiogalactopyranoside (IPTG) induction-based overexpression. Cells were grown at 37 °C until the O.D. reached 0.6–0.8 and then induced using 0.5 mM IPTG and incubated at 18 °C for 16 hr. After induction, the cells were harvested at 7000 rpm for 5 min at 4 °C. Purification was done according to previously published protocols (ref.). Briefly, the proteins were purified by immobilised Ni-NTA-based affinity chromatography in buffer containing 50 mM Tris-HCl pH 8.0, 300 mM NaCl, 0.4% Triton X-100 and 1 mM PMSF. The proteins were eluted using gradient elution between 10 and 250 mM imidazole. Eluted proteins were further purified by size-exclusion chromatography using Superdex 200 in buffer containing 20 mM Tris-HCl pH 8.0, 150 mM KCl. Fractions were pooled, concentrated, and stored at –80 °C until further use.

## Liposome co-sedimentation assay

Liposomes were prepared according to previously published (*Dey et al., 2017*; *Larsson et al., 2020*). Briefly, Lipids 800 nanomoles of POPC (Avanti polar lipids) and 200 nanomoles of DAGs (Sigma) for 20 mole percent, were dissolved in chloroform: methanol (2:1 v/v), mixed in glass tubes and dried to thin film under nitrogen gas stream. Following overnight desiccation, 1 mL of Hydration buffer (50 mM Tris-HCl pH 7.5, 150 mM NaCl, 10 mM $MgCl_2$, 1.7 mM $CaCl_2$, 10 mM β-mercaptoethanol) was added to make 1 mM final lipid concentration. The tubes were vortexed vigorously and subjected to 10 freeze-thaw cycles using liquid nitrogen for 1 min and at 50 °C water bath for 3 min.

C1-GFP (7 nM) was incubated with liposomes for 10 min in 300 µl of hydration buffer. After incubation, the reaction mixture was centrifuged at 100,000xg for 1 hr at 4 °C. The supernatant was collected in a separate tube and both supernatant and pellet were subjected to SDS-PAGE, followed by western blotting using Anti-GFP antibody (Cell Signalling Technology). An in-house linker of 11 amino acids attached with 8xHis and GFP at the C-terminal (DSLEFIASKLA-GFP) was purified and used to probe for GFP as a negative control. Images were captured in the BioRad ChemiDoc imaging system and analysed using ImageJ.

## Western blotting

Overnight grown yeast cells were inoculated in fresh YPD media (10 mL) at 0.2 $OD_{600}$ nm and grown till log phase (2–2.5 $OD_{600}$ nm). Cell pellets were harvested, washed with phosphate buffered saline (PBS) and flash frozen using liquid nitrogen. The pellets were then dissolved in 120 µl of Lysis Buffer 50 mM Tris-HCl [pH 7.5], 150 mM NaCl, 5 mM EDTA, 1% Nonidet P-40/Triton X-100, 1 mM Sodium Pyrophosphate, 1 mM Sodium Orthovanadate, 20 mM NaF, 1 mM Phenylmethylsulfonyl Fluoride, 1% Protease Inhibitor Cocktail (*Nomura and Inoue, 2015*). The dissolved pellets were subjected to lysis with glass beads on a mechanical vortex for 10 cycles of 30 s. The obtained homogenate was centrifuged at 12,000 rpm for 10 min at 4 °C and then the supernatant was boiled with LaemmLi buffer for

15 min. Protein extracts were separated on SDS-PAGE and then transferred on to PVDF membrane (Millipore) using wet transfer at 100 V for 2 hr. Membranes were blocked using 5% BSA in Tris-buffered saline with 0.1% Tween 20 detergent (TBST) and then incubated with appropriate primary antibodies (Anti-Phospho-p44/42 MAPK (Erk1/2) (Thr202/Tyr204) Cell Signalling Technology #9101) to measure pSlt2 and p44/42 MAPK (Erk1/2) Antibody Cell Signalling Technology #9102) (*Sariki et al., 2019*) to measure total Slt2 at 1:5000 dilution or TPI1 at 1:10000 dilution (gift from Dr. Palani Murgan's lab, CSIR-CCMB) followed by washing and incubation with secondary antibody (Anti-rabbit IgG Cell Signalling Technologies #7074 (1:10000). After final wash with TBST, bands were visualised using SuperSignal West Pico PLUS chemiluminescent horseradish peroxidase (HRP) substrate using the BioRad Imaging System. Images were analysed using ImageJ software. For cell wall stress, cells were incubated with CR or CFW for 30 min before harvesting. For cercosporamide treatment, cells were incubated for 30 min at a concentration of 5 µg/mL followed by harvesting and sample preparation.

## Lipidomics sample preparation

Yeast cells were grown overnight in SC media and were diluted in 20 mL media (6x replicates, total 120 mL media for single strain) to 0.2 $OD_{600}$ nm, cultured, and harvested at ~2–2.5 $OD_{600}$ nm. Inhibitor-treated samples were cultured the same way, except that U73122 (0.5 mM and 1 mM) (Sigma), propranolol (1 mM) (Sigma), aureobasidin A (0.025 µg/mL) (Takara) were added while inoculating for the secondary culture at 0.2 OD. For cell wall stress, cells were subjected to CFW (50 µg/mL), 30 min before harvesting. Lipid isolation was performed following modified-Folch method as reported previously (*Abhyankar et al., 2018*; *Kelkar et al., 2019*; *Kumar et al., 2019*; *Mondal et al., 2022*; *Pathak et al., 2018*). Briefly, cell pellets were washed with PBS and flash-frozen in liquid nitrogen. Cells were resuspended in 500 µl of chilled PBS and were lysed using sonicator at 60% amplitude for 1 s ON/ 3 s OFF for 10 cycles with a probe sonicator on ice bed. 10 µl of lysate from each sample was collected for protein estimation required for data normalisation during lipid analysis. 250 µl of PBS was added, followed by addition of $CHCl_3$ and methanol mix with respective internal standards to achieve the ratio of 2:1:1 ($CHCl_3$: Methanol: PBS). Mixture was vortexed thoroughly (two cycles of 30 s) and was phase separated by centrifugation at 3000 RPM at RT. Bottom phase was collected in fresh glass vials followed by addition of 100 µl of formic acid (~10%, vol/vol of lysate volume) to the remaining top phase. After vortexing again, the same way, same volume of $CHCl_3$, as mentioned in previous step, was added for re-extraction by phase separation. Bottom phase was pooled with the already separated lipid in the first round. The total lipids extracted were dried under a stream of nitrogen at room temperature. Protein estimation was performed using Bradford assay reagent (Sigma).

## Lipidome analysis by mass spectrometry

The lipidomic experiment and analysis of the isolated lipid species were performed according to previously described protocols (*Abhyankar et al., 2018*; *Kelkar et al., 2019*; *Kumar et al., 2019*; *Mondal et al., 2022*; *Pathak et al., 2018*). Semi-quantitative analysis of lipids was performed using two mass spectrometers: An Agilent 6545 LC-QTOF (quadrupole-time-of-flight) and a Sciex X500R QTOF employing high-resolution auto MS-MS methods and multiple reaction monitoring high resolution (MRM-HR) scanning, respectively. An Electrospray ionisation (ESI) source was used in both mass spectrometers. The dried lipid extracts were re-solubilised in 200 µL of 2:1 CHCl3: MeOH and 10 µL was injected into the mass spectrometers.

The liquid chromatography separation protocol was the same for both instruments. A Luna C5 column (Phenomenex, 5 µm, 50×4.6 mm) coupled to a C5 guard column (Phenomenex, 4×3 mm) was used for LC separation. The solvents used were buffer A: 95:5 $H_2O$: MeOH +0.1% Formic acid +10 mM ammonium formate and buffer B: 60:35:5 iPrOH: MeOH: $H_2O$+0.1% Formic acid +10 mM ammonium formate. Methods were 30 min long, starting with 0.3 mL/min 100% buffer A for 4 min, 0.5 mL/min linear gradient to 100% buffer B over 14 min, 0.5 mL/min 100% buffer B for 7 min, and equilibration with 0.5 mL/min 100% buffer A for 5 min.

The following settings were used for the ESI-MS positive mode analysis on the Agilent 6545 LC-QTOF: drying gas and sheath gas temperature: 320 °C, drying gas and sheath gas flow rate: 10 L/min, fragmentor voltage: 150 V, capillary voltage: 4000 V, nebuliser (ion source gas) pressure: 45 psi and nozzle voltage: 1000 V. For analysis, a lipid library of DAGs and TAGs was employed in the form

of a Personal Compound Database Library (PCDL), and the peaks were validated based on relative retention times and fragments obtained.

For the Sciex X500R QTOF, the following settings were used for the ESI-MS positive mode analysis: source gas temperature: 400 °C, spray voltage: 4500 V, source gas 1 pressure: 40 psi, source gas 2 pressure: 50 psi. Peaks were quantified using Sciex OS, where the masses of DAGs and TAGs were curated from the lipid maps structural database (LMSD).

All lipid species were quantified by normalising areas under the curve to the protein concentration of the lysate taken at the beginning.

## Live cell microscopy

Endogenous GFP-tagged strains were grown in SC media at 30°C and were harvested between 1.5–2 $OD_{600}$ nm (log phase). MitoTracker Red CMXRos (Invitrogen Inc) was added to a final concentration of 50 nM in 1 mL of liquid cell culture and was incubated further for 30 min with shaking. Cells were then washed five times with PBS and was resuspended in 20 µl of PBS. 3 µl of these cells were immobilised under agar bed on glass slides and visualised under a Zeiss Apotome.2, inverted widefield fluorescence microscope equipped with a HAL 100 illuminator, Plan Apochromat 100x oil objective (NA 1.4), and a AxioCam CCD camera. Images were captured in DIC (Nomarski optics), mCherry (for MitoTracker Red), and FITC fluorescence mode. All images were processed using ImageJ2 software.

## Phylogenetic profiling

Sequences for phylogenetic profiling were collected from a stringent BLAST hit search (*Altschul et al., 1990*) using the protein sequence from *Saccharomyces cerevisiae* as the query. A defined list of organisms was made in order to account for the diversity in the fungal kingdom (Ascomycota, Mucoromycota, Zoopagomycota, Blastocladiomycota, Chytridiomycota, Holomycota) for all the proteins used for coevolution analysis. For the distribution of PKC across tree of life, sequences were collected from the representative organisms from protists to metazoans, consisting of novel, conventional, and atypical PKCs. The domain organisation of the collected sequences was verified in Conserved Domains Database (CDD) (*Lu et al., 2020*) to filter partial sequences. Multiple sequence alignment was then performed using MAFFT, imposing E-INS-i model for multidomain proteins. IQ-TREE (*Nguyen et al., 2015*) was used to generate the phylogenetic tree and iTOL (*Letunic and Bork, 2021*) was used to visualise the tree.

## Coevolution analysis

Interspecies correlation serves as a good indicator to predict functional relationship between proteins. Other than *Dip2* and *PKC1*, a set of conserved housekeeping genes (*GAPDH, PGK1, ACT1, HSP60, CYTC, TPI1, CDC42, RPL5*) and known lipid metabolising enzymes (*Dga1, Dgk1, Lro1, Tgl3, Faa1, Lcb1, Erg4*) were taken as control proteins. The obtained alignment was checked in JalView and arranged alphabetically to maintain homogeneity in comparison. This output alignment file was used to calculate pairwise genetic distances in MEGA X using bootstrap (1000 repetitions), employing the JTT model (Jones-Taylor Thornton model) and uniform mutation rates that represent the number of substitutions per site between two homologous proteins.

The obtained genetic distances were exported as a distance matrix and Pearson correlation coefficient was calculated using the following equation-

$$r = \frac{\sum\limits_{i=1}^{n} \left(R_i - \bar{R}\right)\left(S_i - \bar{S}\right)}{\sqrt{\sum\limits_{i=1}^{n}\left(R_i - \bar{R}\right)^2}\sqrt{\sum\limits_{i=1}^{n}\left(S_i - \bar{S}\right)^2}}$$

The correlation coefficients were converted to normally distributed metric using Fisher's *r* to *z* transformation,

$$r' = \frac{1}{2} ln \left| \frac{1 + r}{1 - r} \right|$$

where,

*r*=Pearson correlation coefficient
*r'*=Fisher-transformed correlation coefficient
These transformed coefficients (r') were further compared to generate z score/z test statistic using the given equation,

$$z = \frac{r'_1 - r'_2}{\sqrt{\frac{1}{N_1 - 3} + \frac{1}{N_2 - 3}}}$$

where,
r'$_1$ is the first Fisher-transformed correlation coefficient
r'$_2$ is the second Fisher-transformed correlation coefficient
$N_1$ and $N_2$ denote the number of common organisms in the first and second correlation, respectively. z-scores were calculated for all pairs where r'$_1$, representing Dip2-PKC correlation coefficient and r'$_2$ representing correlation coefficient for Dip2 with other protein controls, giving negative values for higher correlations and positive for lower ones. p-values were calculated from the obtained z-scores.

Patristic distance is the sum of the lengths of the branches that link two terminal nodes in a tree, denoting the divergence between the two nodes. Using the same collected sequences, phylogenetic trees were constructed by standard methods- Bayesian Inference, Maximum Likelihood (using JTT model), Neighbor Joining (NJ) and Minimum Evolution. Patristic distances were calculated for each phylogenetic model using PATRISTIC software and correlation coefficients were generated from the obtained distance matrices. Similar phylogenetic patterns were observed where the homogeneity in correlation coefficient is maintained. Basically, a higher correlation score represents a stronger relationship between the rate at which the two proteins have evolved along the multiple branches of their phylogenetic trees.

## Statistical analysis

All the statistical analyses were performed in Microsoft Excel and GraphPad Prism 9.3.1 (471). Statistical analysis of the differences between two groups was performed using a two-tailed, unpaired and parametric, Student's t-test. Error bars represent standard deviations (SD), except for lipidomic analyses, where error bars are plotted as standard error of mean (SEM). Significance of differences are marked based on the p-value obtained. *$p < 0.05$; **$p < 0.01$; ***$p < 0.001$; ****$p < 0.0001$; ns, not significant. All the experiments are replicated minimally thrice with technical replicates.

## Acknowledgements

We thank Dr. Krishnaveni Mishra (University of Hyderabad) and Dr. Purusharth I Rajyaguru (Indian Institute of Science) for sharing yeast knockout strains with us. We thank Dr. Kaustuv Sanyal and Kuladeep Das (Jawaharlal Nehru Centre for Advanced Scientific Research) for generating *Δpsi1Δdip2* strain for us. We thank Dr. Sriram Varahan (CSIR-CCMB) for engaging discussions and sharing resources. We thank Saddam Shekh and Aakash Chandramouli for technical assistance with the LCMS experiments at IISER Pune. SS thanks University Grants Commission, India; SM and AC thank Council of Scientific and Industrial Research (CSIR), India, for the research fellowship. RSN thanks healthcare theme projects- Fundamental and Innovative CSIR in Science of Tomorrow (FIRST; MLP-0162) and Niche Creation Project (NCP; MLP-0138) of CSIR, India; JC Bose Fellowship of SERB, India; and Centre of Excellence Project of Department of Biotechnology, India. SSK thanks Swarnajayanti Fellowship from the Science and Engineering Research Board (SERB), Department of Science and Technology (DST), Government of India (Grant number SB/SJF/2021-22/01) and the Department of Science & Technology–Funds for Improvement of S&T Infrastructure Development (DST-FIST) (grant number SR/FST/LSII-043/2016) to the Department of Biology, IISER Pune for setting up the Biological Mass Spectrometry Facility.

# Additional information

## Competing interests

Rajan Sankaranarayanan: Reviewing editor, eLife. The other authors declare that no competing interests exist.

## Funding

| Funder | Grant reference number | Author |
|---|---|---|
| Science and Engineering Research Board | Swarnajayanti fellowship SB/SJF/2021-22/01 | Siddhesh S Kamat |
| Council of Scientific and Industrial Research | Fundamental and innovative CSIR in science of tomorrow, Niche creation project FIRST; MLP-0162 | Rajan Sankaranarayanan |
| Niche creation project | NCP; MLP-0138 | Rajan Sankaranarayanan |
| Department of Science & Technology | Funds for Improvement of S&T Infrastructure Development (DST-FIST) SR/FST/LSII-043/2016 | Siddhesh S Kamat |
| University Grants Commission India | | Sakshi Shambhavi |
| Council for Scientific and Industrial Research | | Sudipta Mondal Arnab Chakraborty |
| Science and Engineering Research Board | JC Bose Fellowship of SERB | Rajan Sankaranarayanan |
| Centre of Excellence Project of Department of Biotechnology, India | | Rajan Sankaranarayanan |
| Department of Science & Technology | SR/FST/LSII- 043/2016 | Siddhesh S Kamat |

The funders had no role in study design, data collection and interpretation, or the decision to submit the work for publication.

## Author contributions

Sakshi Shambhavi, Sudipta Mondal, Conceptualization, Data curation, Formal analysis, Validation, Investigation, Visualization, Methodology, Writing – original draft, Writing – review and editing; Arnab Chakraborty, Data curation, Formal analysis, Validation, Investigation, Visualization, Methodology, Writing – review and editing; Nikita Shukla, Bapin Kumar Panda, Data curation, Formal analysis, Validation, Visualization, Methodology, Writing – review and editing; Santhosh Kumar, Data curation, Validation, Visualization, Methodology, Writing – review and editing; Priyadarshan Kinatukara, Conceptualization, Validation, Visualization, Writing – review and editing; Biswajit Pal, Validation, Visualization, Project administration, Writing – review and editing; Siddhesh S Kamat, Resources, Data curation, Software, Formal analysis, Supervision, Funding acquisition, Validation, Investigation, Visualization, Methodology, Writing – review and editing; Rajan Sankaranarayanan, Conceptualization, Resources, Software, Supervision, Funding acquisition, Validation, Investigation, Writing – original draft, Project administration, Writing – review and editing

## Author ORCIDs

Sakshi Shambhavi (iD) https://orcid.org/0000-0002-8852-1542
Bapin Kumar Panda (iD) https://orcid.org/0009-0007-7984-2241
Rajan Sankaranarayanan (iD) https://orcid.org/0000-0003-4524-9953

Reviewer #1 (Public review): https://doi.org/10.7554/eLife.104011.3.sa1
Reviewer #2 (Public review): https://doi.org/10.7554/eLife.104011.3.sa2

Author response https://doi.org/10.7554/eLife.104011.3.sa3

## Additional files

### Supplementary files
Supplementary file 1. List of yeast strains.

Supplementary file 2. List of Primers.

Supplementary file 3. List of plasmids.

Supplementary file 4. Software and algorithms.

Supplementary file 5. Reagents or Resources.

MDAR checklist

### Data availability
All data generated or analysed during this study are included in the manuscript and supporting files.

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
