## [Editor Report · eLife Assessment]

This is an interesting study that adds **useful** new data addressing how different DAG pools influence cellular signaling. The study dissects how the enzyme Dip2 modulates the minor lipid signaling DAG pool, which is distinct from the lipid metabolism DAG pool utilized in membrane production. Overall the analysis is **solid** and broadly supports the claims.

---

## [Referee Report · Reviewer #1 (Public review)]

Summary:

The study dissects distinct pools of diacylglycerol (DAG), continuing a line of research on the central concept that there is a major lipid metabolism DAG pool in cells, but also a smaller signaling DAG pool. It tests the hypothesis that the second pool is regulated by Dip2, which influences Pkc1 signaling. The group shows that stressed yeast increase specific DAG species C36:0 and 36:1, and propose this promotes Pkc1 activation via Pck1 binding 36:0. The study also examines how perturbing the lipid metabolism DAG pool via various deletions such as lro1, dga1, and pah1 deletion impacts DAG and stress signaling. Overall this is an interesting study that adds new data to how different DAG pools influence cellular signaling.

Strengths:

The study nicely combined lipidomic profiling with stress signaling biochemistry and yeast growth assays.

Weaknesses:

One suggestion to improve the study is to examine the spatial organization of Dip2 within cells, and how this impacts its ability to modulate DAG pools. Dip2 has previously been proposed to function at mitochondria-vacuole contacts (Mondal 2022). Examining how Dip2 localization is impacted when different DAG pools are manipulated such as by deletion Pah1 (also suggested to work at yeast contact sites such as the nucleus-vacuole junction), or with Lro1 or Dga1 deletion would broaden the scope of the study.

Comments on revisions:

The revision addresses several of the concerns raised previously. Most importantly, it softens several conclusions that more clearly delineates limitations of the study. The study has yet to address how Dip2 and Pkc1 crosstalk, but new text addresses this limitation. There is also more analysis of Dip2 localization in other conditions where cell DAG pools are elevated (ie a LRO1 and DGA1 double KO, as well as PAH1 KO). Loss of these proteins elevates ER DAG, but Dip2 remains mitochondrially associated. This may imply DAG specificity, or that changes to DAG pools globally does not impact Dip2 import into mitochondria.

---

## [Referee Report · Reviewer #2 (Public review)]

Summary:

The authors use yeast genetics, lipidomic and biochemical approaches to demonstrate the DAG isoforms (36:0 and 36:1) can specifically activate PKC. Further, these DAG isoforms originate from PI and PI(4,5)P2. The authors propose that the Psi1-Plc1-Dip2 functions to maintain a normal level of specific DAG species to modulate PKC signalling.

Strengths:

Data from yeast genetics are clear and strong. The concept is potentially interesting and novel.

Weaknesses: More evidence is needed to support the central hypothesis. The authors may consider the following:

(1) Figure 2: the authors should show/examine C36:1 DAG. Also, some structural evidence would be highly useful here. What is the structural basis for the assertion that the PKC C1 domain can only be activated by C36:0/1 DAG but not other DAGs? This is a critical conclusion of this work and clear evidence is needed.

(2) Does Dip2 colocalize with Plc1 or Pkc1? Does Dip2 reach the plasma membrane upon Plc activation?

Comments on revisions:

The authors have addressed my concerns.

---

## [Author Response]

The following is the authors’ response to the original reviews

**Public Reviews:**

**Reviewer #1 (Public review):**
Summary:The study dissects distinct pools of diacylglycerol (DAG), continuing a line of research on the central concept that there is a major lipid metabolism DAG pool in cells, but also a smaller signaling DAG pool. It tests the hypothesis that the second pool is regulated by Dip2, which influences Pkc1 signaling. The group shows that stressed yeast increase specific DAG species C36:0 and 36:1, and propose this promotes Pkc1 activation via Pck1 binding 36:0. The study also examines how perturbing the lipid metabolism DAG pool via various deletions such as lro1, dga1, and pah1 deletion impacts DAG and stress signaling. Overall this is an interesting study that adds new data to how different DAG pools influence cellular signaling.Strengths:The study nicely combined lipidomic profiling with stress signaling biochemistry and yeast growth assays.

We thank the reviewer for finding this study of interest and appreciating our multi-pronged approach to prove our hypothesis that a distinct pool of DAGs regulated by Dip2 activate PKC signalling.

Weaknesses:One suggestion to improve the study is to examine the spatial organization of Dip2 within cells, and how this impacts its ability to modulate DAG pools. Dip2 has previously been proposed to function at mitochondria-vacuole contacts (Mondal 2022). Examining how Dip2 localization is impacted when different DAG pools are manipulated such as by deletion Pah1 (also suggested to work at yeast contact sites such as the nucleus-vacuole junction), or with Lro1 or Dga1 deletion would broaden the scope of the study.

We thank the reviewer for the suggestion to trace the localization of Dip2 in the absence of various DAG-acting enzymes. To address this, we generated Dip2-GFP knock-in (KI) in *Δpah1, Δlro1* and *Δdga1* strains, confirming successful integration by western blotting using an anti-GFP antibody. We then performed microscopy to examine the localization of Dip2. Since Dip2 is a mitochondria-vacuole contact site protein that predominantly localizes to mitochondria (approximately 60% puncta of Dip2 localize to mitochondria) (Mondal et al. 2022), we co-stained the cells with MitoTracker red to visualize mitochondria.

Consistent with our previous findings, Dip2 colocalizes with the MitoTracker red in *WT* (Figure 3-figure supplement 2 A). As suggested by the reviewer, we deleted *PAH1*, which converts phosphatidic acid to DAGs and is also known to work at the nucleus-vacuole junction. On examining whether absence of PAH1 influences the localization of Dip2, we found that there is no change in Dip2’s spatial organization. This could also be due to no observable change in the DAG species on deleting PAH1, as noted in our lipidomic studies (Figure 4. figure supplement 2A). These observations suggest that in a homeostatic condition, Pah1 does not affect the DAG pool acted upon by Dip2 and therefore has no influence on Dip2’s subcellular localization. This data has been incorporated in the revised manuscript (line no. 286-289) and Figure 4-figure supplement 2D-E.

Similarly, we probed for the localization of Dip2 in *LRO1* and *DGA1* knock out strains. These enzymes are responsible for converting bulk DAGs to TAGs. We have previously shown that Dip2 is selective for only C36:0 and C36:1 and does not act on the bulk DAGs (Mondal et al. 2022). Both Lro1 and Dga1 are endoplasmic reticulum (ER) resident proteins and the bulk DAG accumulation in their knockouts is shown to be in the ER (Li et al. 2020), not influencing the mitochondrial DAG pool. On tracing Dip2’s localization in these knockouts, we found that Dip2 remains in the mitochondria (Figure 3-figure supplement 2, Figure 4. figure supplement 2D,E). These results suggest that Dip2 localization is not influenced by bulk DAG accumulation, reinforcing its specificity toward selective DAGs, which are likely to be present at mitochondria and mitochondria-vacuole contact sites. We have added this data in the revised manuscript (line no. 240-246) with Figure 3. figure supplement 2.

**Reviewer #2 (Public review):**
Summary:The authors use yeast genetics, lipidomic and biochemical approaches to demonstrate the DAG isoforms (36:0 and 36:1) can specifically activate PKC. Further, these DAG isoforms originate from PI and PI(4,5)P2. The authors propose that the Psi1-Plc1-Dip2 functions to maintain a normal level of specific DAG species to modulate PKC signalling.Strengths:Data from yeast genetics are clear and strong. The concept is potentially interesting and novel.

We would like to thank the reviewer for the positive comments on our work and finding the study novel and interesting.

Weaknesses:More evidence is needed to support the central hypothesis. The authors may consider the following:(1) Figure 2: the authors should show/examine C36:1 DAG. Also, some structural evidence would be highly useful here. What is the structural basis for the assertion that the PKC C1 domain can only be activated by C36:0/1 DAG but not other DAGs? This is a critical conclusion of this work and clear evidence is needed.

We thank the reviewer for the insightful comments. We were unable to include C36:1 DAG in our in vitro DAG binding assays because it is not commercially available. We have now explicitly mentioned it in the revised manuscript (Line no. 186).

We agree with the reviewer that PKC activated by C36:0 and C36:1 DAGs is a critical conclusion of our work. While we understand that there is no obvious structural explanation as to how the DAG binding C1 domain of PKC attains the acyl chain specificity for DAGs, our conclusion that yeast Pkc1 is selective for C36:0 and C36:1 DAGs, is supported by a combination of robust in vitro and in vivo data:

(1) In Vitro Evidence: The liposome binding assays demonstrate that the Pkc1 C1 domain binds only to the selective DAG and does not interact with bulk DAGs.

(2) In Vivo Evidence: Lipidomic analyses of wild-type cells subjected to cell wall stress reveal increased levels of C36:0 and C36:1 DAGs, while levels of bulk DAGs remain unaffected.

These findings collectively indicate that Pkc1 neither binds nor is activated by bulk DAGs, reinforcing its specificity for C36:0 and C36:1 DAGs.

Moreover, the structural basis of this selectivity would require either a specific DAG-bound C1 domain structure of Pkc1, which is difficult owing to the flexibility of the longer acyl chains present in C36:0 and C36:1 DAGs. In addition, capturing the full-length Pkc1 structure that might provide deeper insights has been challenging for several other groups. Also, we hypothesize that the DAG selectivity by Pkc1 is more of a membrane phenomenon wherein these DAGs might create a specific microdomain or form a particular curvature that is sensed by Pkc1. Investigating this would require extensive structural and biophysical studies, that are beyond the scope of the current work but are planned for future research.

(2) Does Dip2 colocalize with Plc1 or Pkc1?

As shown in our previous study (Mondal et al. 2022) and in the above section (Figure 3. figure supplement 2(A-B)), Dip2 predominantly localizes to the mitochondria. Pkc1, on the other hand, is known to be found in the cytosol, plasma membrane and bud site (Andrews and Stark 2000). We also checked the localization of Pkc1, co-stained with mitotracker-red and observed no significant overlap between the two, confirming that Pkc1 does not colocalize with Dip2 (Author response image 1).

**Author response image 1. sa3fig1:** Live cell microscopy for tracing Pkc1 localization. (**A**) Microscopy image panel showing DIC image (left), fluorescence for (A) Pkc1 tagged with GFP, mitotracker-red for staining mitochondria and the merged image for both the fluorophores (right). Scale bar represents 5 µm. (**B**) Line scan plotted for the fluorescence intensity of Pkc1-GFP along with mitotracker-red across the line shown in the merged panel.

Moreover, as suggested by the reviewer, we also checked the localization of Plc1 and found that Plc1 is present in cytosol and shows a partial colocalization with the mitochondria (Figure 4-figure supplement 3A-B). As some puncta of Dip2 also colocalize with the vacuoles, we checked whether Plc1 also follows such localization pattern. We costained Plc1-GFP with FM4-64, a vacuolar membrane dye and observed that Plc1 partially localizes to vacuoles as well (Figure 4-figure supplement 3C-D). This is also observed in a previous study where Plc1 was found in a subcellular fractionation of isolated yeast vacuoles and total cell lysate (Jun, Fratti, and Wickner 2004). We also checked similar to Dip2, whether Plc1 also localizes to the Mitochondria-vacuole contact site by using tri-colour imaging with FM4-64 for vacuole, DAPI for mitochondria and GFP tagged Plc1. We were not able to trace Dip2 and Plc1 simultaneously as we could not generate a strain endogenously tagged with two different colours even after several attempts. However, from our observations, we can conclude that Plc1 partially localizes to mitochondria and vacuole and might be locally producing the selective DAGs to be acted upon by Dip2. We have incorporated this data in the revised manuscript (line no. 301-304) with Figure 4-figure supplement 3.

For probing the localization of Dip2 upon Plc1 activation, we used cell wall stress- a condition inducing Plc1 activation for selective DAG production (this study). Under this condition, we probed the localization of Dip2 by fluorescent microscopy and found that Dip2 does not move to the plasma membrane but remains localized to mitochondria (Figure. 1. figure supplement 3). This result has been added in the revised manuscript (line no. 153-160) with Figure. 1-figure supplement 3.

This raises intriguing questions regarding the spatial regulation of Pkc1 by Dip2. Since Dip2’s localization remains unaffected, whether the selective DAGs, presumably at the mitochondria, move to the plasma membrane for Pkc1 activation or the Pkc1 translocates to the mitochondria needs further exploration. Addressing these possibilities will require a combination of genetic approaches, organellar lipidomics, and advanced microscopy, which we aim to explore in future studies.

References:

Andrews, P. D., and M. J. Stark. 2000. “Dynamic, Rho1p-Dependent Localization of Pkc1p to Sites of Polarized Growth.” Journal of Cell Science 113 (Pt 15): 2685–93. doi:10.1242/jcs.113.15.2685.

Jun, Youngsoo, Rutilio A. Fratti, and William Wickner. 2004. “Diacylglycerol and Its Formation by Phospholipase C Regulate Rab- and SNARE-Dependent Yeast Vacuole Fusion*.” Journal of Biological Chemistry 279(51): 53186–95. doi:10.1074/jbc.M411363200.

Li, Dan, Shu-Gao Yang, Cheng-Wen He, Zheng-Tan Zhang, Yongheng Liang, Hui Li, Jing Zhu, et al. 2020. “Excess Diacylglycerol at the Endoplasmic Reticulum Disrupts Endomembrane Homeostasis and Autophagy.” BMC Biology 18(1): 107. doi:10.1186/s12915-020-00837-w.

Mondal, Sudipta, Priyadarshan Kinatukara, Shubham Singh, Sakshi Shambhavi, Gajanan S Patil, Noopur Dubey, Salam Herojeet Singh, et al. 2022. “DIP2 Is a Unique Regulator of Diacylglycerol Lipid Homeostasis in Eukaryotes.” eLife 11: e77665. doi:10.7554/eLife.77665.